# Neurons along the auditory pathway exhibit a hierarchical organization of prediction error

Gloria G. Parras [1,2], Javier Nieto-Diego[1,2], Guillermo V. Carbajal [1,2], Catalina Valdés-Baizabal[1,2], Carles Escera [3,4,5] & Manuel S. Malmierca [1,2,6]

Perception is characterized by a reciprocal exchange of predictions and prediction error signals between neural regions. However, the relationship between such sensory mismatch responses and hierarchical predictive processing has not yet been demonstrated at the neuronal level in the auditory pathway. We recorded single-neuron activity from different auditory centers in anaesthetized rats and awake mice while animals were played a sequence of sounds, designed to separate the responses due to prediction error from those due to adaptation effects. Here we report that prediction error is organized hierarchically along the central auditory pathway. These prediction error signals are detectable in subcortical regions and increase as the signals move towards auditory cortex, which in turn demonstrates a large-scale mismatch potential. Finally, the predictive activity of single auditory neurons underlies automatic deviance detection at subcortical levels of processing. These results demonstrate that prediction error is a fundamental component of singly auditory neuron responses.

[1] Auditory Neuroscience Laboratory, Institute of Neuroscience of Castilla y León (INCYL), Salamanca, 37007 Castilla y León, Spain. [2] The Salamanca Institute for Biomedical Research (IBSAL), Salamanca, 37007 Castilla y León, Spain. [3] Brainlab-Cognitive Neuroscience Research Group, Department of Clinical Psychology and Psychobiology, University of Barcelona, Barcelona, 08035 Catalonia, Spain. [4] Institute of Neurosciences, University of Barcelona, Barcelona, 08035 Catalonia, Spain. [5] Institut de Recerca Sant Joan de Déu, Esplugues de Llobregat, 08950 Catalonia, Spain. [6] Department of Cell Biology and Pathology, Faculty of Medicine, University of Salamanca, Salamanca, 37007 Castilla y León, Spain. Gloria G. Parras and Javier Nieto-Diego contributed equally to this work. Correspondence and requests for materials should be addressed to M.S.M. (email: msm@usal.es)

Unexpected events tend to convey relevant information, making their prompt detection fundamental for survival[1]. Brain responses to a perceptual mismatch between expected and actual sensory inputs have been extensively recorded in all sensory systems, including auditory, visual, somatosensory and olfactory modalities[2, 3]. In the case of audition, these responses are thought to underlie the brain's ability to identify what sounds or auditory objects are[4], suggesting that they may be a key feature of perceptual processing[3, 5]. Auditory prediction errors can be induced using oddball sequences[5], in which a repetitive (standard) tone is replaced randomly by a different (deviant) tone with a low probability. Neural responses, recorded from the human scalp with electroencephalography while people heard such oddball stimuli, have revealed a characteristic pattern of activity, the so-called mismatch negativity (MMN) response[6].

The MMN response is widely considered to represent a prediction error signal, a member of a hierarchy of prediction errors[3, 7, 8]. Hierarchical predictive coding is a neurobiologically informed theory of general brain function[9, 10] that unifies many concepts and experimental evidence about perceptual systems into a common framework. According to this framework, cortical processing stations send predictions to lower hierarchical levels to aid the suppression of any ascending neuronal activity evoked by sensory events that can be anticipated. These stations also forward prediction errors to higher hierarchical levels whenever their current predictions fail. This framework explains both repetition suppression, or response attenuation with stimulus repetition[11, 12], and deviance detection, or automatic enhancement of responses to sensory inputs that deviate from a strong prediction[13, 14]. Because it encompasses these different facets, the main concepts of predictive coding have been used to describe a variety of brain responses and brain dynamics, including the MMN[3, 4, 13]. Thus, it is now widely accepted that large-scale mismatch responses such as those seen in humans or animals listening to an auditory oddball stimulus[15, 16], reflect the predictive activity of the auditory and other sensory systems[3, 7]. These responses can be seen even at early processing stages[8], including subcortical midbrain and thalamus[2].

However, at the cellular level, such mismatch responses could also arise from a simpler neurophysiological mechanism[17, 18], namely, stimulus-specific adaptation (SSA)[19, 20], which is response decrement to a stimulus repetition[1] that leaves neuronal responses to novel stimuli almost unaffected. SSA is a widespread property of auditory neurons, increasing from midbrain[21–24] through the thalamus[25] to primary[26, 27] and non-primary[27] auditory cortices, and is assumed to be due to synaptic depression[2, 26]. Due to SSA, single neuron responses along the auditory pathway show a differential response to standard (highly repeated sounds) and deviant (low probability sounds) tones under oddball stimulation, thereby resembling a cellular version of the MMN but at the neuronal level[2, 19]. This similarity has caused some researchers to suggest that SSA is all the brain needs to generate the MMN[17]. Yet, this theory does not take into account predictive activity in single neurons, which has been demonstrated in different contexts and systems. Single neurons in primary auditory cortex have also been probed for predictive activity[15, 19], but the results, have been controversial[18]: Some studies did not find evidence for deviance detection[28, 29], while others found similar results but interpreted them differently and suggested that auditory cortical neurons do detect deviance[26]. Only one recent study in mouse primary auditory cortex explicitly showed deviance detection in late responses of layer II/III excitatory cortical neurons[30].

Auditory signals follow an ascending pathway and are interrupted at least three times: at the cochlear nuclei, the superior olivary complex, the nuclei of the lateral lemniscus, and the inferior colliculus. The different nuclei in these structures encode specific features of the acoustic stimulus. However, the system is even more complex, as the ascending auditory pathway can actually be divided into two broad categories of parallel processing stations. These have been referred to as the "lemniscal line system" and "lemniscal adjunct system"[31] and have been identified in both the auditory (referring to the lateral lemniscus) and somatosensory systems (referring to the medial lemniscus). Currently, the terms "lemniscal" and "non-lemniscal" are widely used to refer to two general categories of pathways between the IC and the forebrain[32, 33]. Neurons in the lemniscal areas of the auditory system ("cochleotopic" or core areas) tend to be sharply tuned and tonotopically organized, whereas neurons in the non-lemniscal areas ("diffuse" or belt areas) are broadly tuned and tonotopy is not evident. In general, the lemniscal part of the inferior colliculus projects to the lemniscal part of the auditory thalamus, which projects to the core or primary auditory cortex, and the non-lemniscal inferior colliculus projects to the non-lemniscal areas of the auditory thalamus, which project to the non-primary or belt areas of auditory cortex[34].

In this study, we recorded the individual responses of subcortical and cortical neurons along the auditory pathway while anaesthetized rats and awake mice were played a recently-developed auditory oddball sequences, which are designed to separate repetition suppression from prediction error[35]. We report the data from a large sample of anesthetized rats and from a smaller sample of awake mice to assess the generalizability of any findings across rodent species and arousal states. Our data show that differential responses to deviant and standard tones in oddball sequences indeed reflect active predictive activity and not simply SSA in single neurons, and that this predictive activity follows a hierarchical pattern that extends to subcortical structures. These results unify three coexisting views of perceptual deviance detection at different levels of description: neuronal physiology, cognitive neuroscience and the theoretical predictive coding framework.

## Results

**Evidence of prediction error in single auditory neurons.** The goal of the present experiments was to test responses of single neurons of the central auditory system of the rat for signs of predictive activity under oddball stimulation. We recorded extracellular single neuron activity in response to sinusoidal tones in different auditory centers of the rat brain (Fig. 1a). Rats were deeply anesthetized prior to surgery preparation and during the whole recording session. One single neuron was recorded at a time, using one tungsten electrode inserted into the brain, and local field potential (LFP) activity was simultaneously recorded from the same electrode.

The predictive coding framework assumes that the generation of both predictions and prediction errors takes place at every hierarchical level of a sensory system[10]. In principle, this assumption could include subcortical processing stations[12]. Unfortunately, there is little evidence supporting this possibility, since most previous research on predictive brain activity was focused on cortical responses[7, 8]. In order to collect a representative sample from different processing stations along the auditory pathway, we recorded a total of 210 neurons (Table 1) from the following: the auditory midbrain—specifically the inferior colliculus (IC), the auditory thalamus—specifically the medial geniculate body (MGB), and the auditory cortex (AC) of anesthetized rats while the animal was played sequences of pure tones (Fig. 1b). According to the well-established functional and anatomical organization of the auditory system[34], recorded neurons in the IC, MGB and AC were grouped as lemniscal (L) or

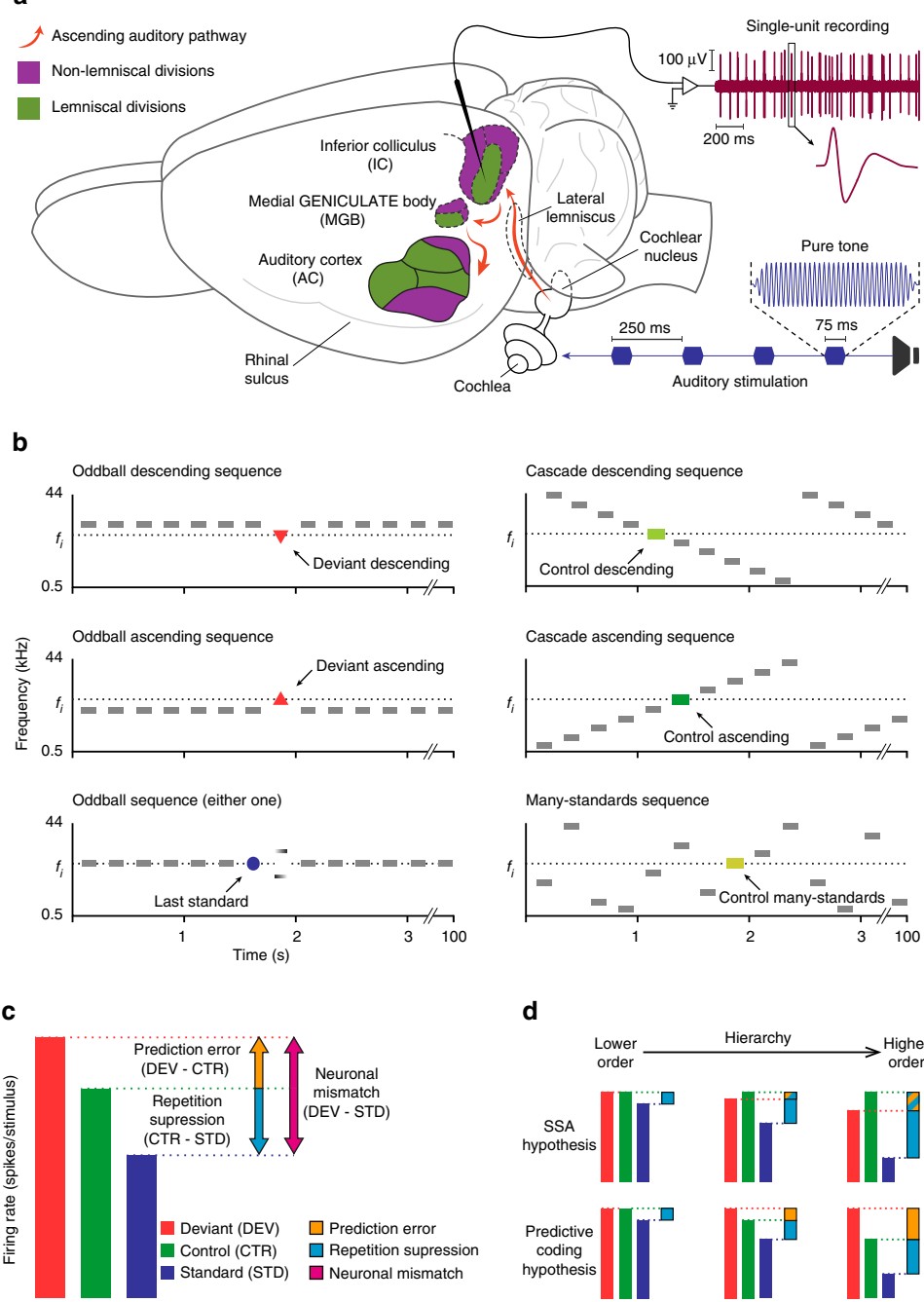

**Fig. 1** Experimental design. **a** Sketch of experimental setup. While stimulating with sequences of pure tones, isolated neurons were recorded from three auditory nuclei of anesthetized rats: IC, MGB and AC (colored). The schematic representation of the ascending auditory pathway information flow (orange) shows how lemniscal (green) and non-lemniscal (purple) subdivisions can be distinguished in the IC, MGB and AC. **b** Stimulation sequences. For each neuron, 10 tones of evenly-spaced frequencies were selected to construct the stimulation sequences. Each tone $f_i$ (i = 1…10) lying inside the neuron's receptive field could be presented in two experimental conditions (deviant and standard, in separated oddball sequences, left column), and two control conditions (cascade and many-standards, right column) for adaptation effects. Note that ascending and descending deviant tones will be compared to the control ascending or control descending sequences, respectively. They will also be compared to the many-standards sequence for both types of deviants (see Methods). **c** Decomposition of neuronal mismatch responses (DEV−STD) to the oddball sequence using either one of the control conditions. Under the assumption of predictive coding, CTR−STD (if positive) represents repetition suppression, and DEV−CTR (if positive) represents prediction error. **d** Two hypothetical scenarios according to two possible competing mechanisms accounting for the neuronal mismatch: SSA (top) and predictive coding (bottom). For SSA, there is response suppression to the standard (blue bars), which progressively increases from lower order to higher order. In addition, due to suppression of the deviant relative to control, the prediction error is increasingly negative (blue and orange bars) as one progresses to higher-order regions. For predictive coding, repetition suppression of the standard (blue bars) increases from lower to higher-order regions. Unlike SSA, responses to deviants are higher than controls, especially in higher-order regions, leading to a positive prediction error (orange bars)

**Table 1 Summary of principal urethane data set**

|  | IC$_L$ | IC$_{NL}$ | MGB$_L$ | MGB$_{NL}$ | AC$_L$ | AC$_{NL}$ |
|---|---|---|---|---|---|---|
| Neurons | 26 | 56 | 25 | 33 | 34 | 36 |
| Points/required | 149/104 | 523/401 | 79/69 | 211/153 | 250/125 | 307/29 |
| DEV (spikes) | 1.37 | 0.99 | 0.72 | 0.69 | 0.95 | 0.98 |
| STD (spikes) | 1.25 | 0.22 | 0.20 | 0.16 | 0.24 | 0.21 |
| Cascade (spikes) | 1.66 | 0.97 | 0.74 | 0.57 | 0.77 | 0.59 |
| Many-standards (spikes) | 1.91 | 0.95 | 0.90 | 0.65 | 0.85 | 0.52 |
| Spike count differences |  |  |  |  |  |  |
| DEV−STD | 0.12 | 0.78 | 0.52 | 0.54 | 0.71 | 0.76 |
| *p* value | 0.000 | 0.000 | 0.000 | 0.000 | 0.000 | 0.000 |
| Cascade−STD | 0.41 | 0.76 | 0.53 | 0.42 | 0.53 | 0.38 |
| *p* value | 0.000 | 0.000 | 0.000 | 0.000 | 0.000 | 0.000 |
| DEV−Cascade | −0.29 | **0.02** | −0.01 | **0.12** | **0.18** | **0.38** |
| *p* value | 0.000 | 0.024 | 0.021 | 0.019 | 0.017 | 0.000 |
| Many-standards−STD | 0.57 | 0.73 | 0.70 | 0.50 | 0.60 | 0.31 |
| *p* value | 0.003 | 0.000 | 0.000 | 0.000 | 0.000 | 0.000 |
| DEV−Many-standards | 0.04 | 0.04 | −0.26 | 0.03 | **0.11** | **0.46** |
| *p* value | 0.190 | 0.155 | 0.003 | 0.671 | 0.049 | 0.000 |
| Differences using Cascade controls |  |  |  |  |  |  |
| iMM | 0.14 | 0.49 | 0.34 | 0.52 | 0.50 | 0.60 |
| *p* value | 0.000 | 0.000 | 0.000 | 0.000 | 0.000 | 0.000 |
| iRS | 0.22 | 0.46 | 0.46 | 0.46 | 0.39 | 0.33 |
| *p* value | 0.000 | 0.000 | 0.000 | 0.000 | 0.000 | 0.000 |
| iPE | −0.08 | **0.03** | −0.12 | **0.06** | **0.11** | **0.27** |
| *p* value | 0.000 | 0.024 | 0.021 | 0.019 | 0.017 | 0.000 |
| Differences using Many-standards |  |  |  |  |  |  |
| iMM | 0.14 | 0.48 | 0.30 | 0.50 | 0.50 | 0.61 |
| *p* value | 0.000 | 0.000 | 0.000 | 0.000 | 0.000 | 0.000 |
| iRS | 0.16 | 0.46 | 0.44 | 0.49 | 0.43 | 0.34 |
| *p* value | 0.003 | 0.000 | 0.000 | 0.000 | 0.000 | 0.000 |
| iPE | −0.02 | 0.02 | −0.14 | 0.01 | **0.07** | **0.27** |
| *p* value | 0.190 | 0.155 | 0.003 | 0.671 | 0.049 | 0.000 |

For each auditory station: Number of recorded neurons and tested neuron/tone combinations (points), along with estimated minimum sample size (of points) required for a statistical power of 0.8 (see Methods subsection on 'Statistical Analyses'). Median values for baseline-corrected spike counts (spikes) to the different conditions. Median differences between the former measures, and associated *p* values against zero (Friedman test with post hoc multiple comparison, Fisher's Least Significant Difference method, uncorrected for 6 independent tests). All *p* values are rounded to 3 decimal figures, so a value of 0.000 means "*p* < 0.0005". Median indices of neuronal mismatch (iMM), repetition suppression (iRS) and prediction error (iPE), computed from each of the two control sequences (cascade or many-standards), and their corresponding *p* values (note that *p* values are the same for absolute differences and normalized indices, since these indices are median differences between normalized responses, and the non-parametric test is independent of scaling). Values related to predictive neuronal activity are highlighted in bold case, since they represent the most significant result of this research

non-lemniscal (NL)[2, 27, 34], thus leading to six different processing stations. These included the following: (1) the central nucleus of the IC, i.e., the lemniscal division of the IC (IC$_L$); (2) the dorsal, lateral, and rostral cortices of the IC, i.e., the non-lemniscal divisions of the IC (IC$_{NL}$); (3) the ventral division of the MGB, i.e., the lemniscal division of the auditory thalamus, (MGB$_L$); (4) the medial and dorsal divisions of the MGB, i.e., the non-lemniscal regions of the MGB (MGB$_{NL}$); (5) the primary auditory cortical fields: primary, anterior, and ventral auditory fields, which collectively constitute the core or lemniscal AC (AC$_L$), and finally, (6) the posterior and the suprarhinal auditory field, which together form the belt or non-lemniscal division of the AC (AC$_{NL}$; for a full list of abbreviations, refer to Supplementary Table 1; Fig. 2; see Methods section).

For each recorded neuron, we presented a set of oddball sequences, using tones selected from the neuron's frequency-response area, and we computed a "neuronal mismatch response" as the difference between responses to deviant (DEV) and standard (STD) conditions for each tone (Fig. 1c). To determine whether this difference (usually DEV > STD) reflected predictive activity, instead of (or in addition to) SSA, we also presented two cascaded sequences (ascending and descending) and one many-standards sequence as controls[35, 36] (Fig. 1b). These latter sequences contained all tones used in oddball sequences (see Methods section). The main rationale behind this design is that, in the control conditions, each tone has the same low (10%)

probability of occurrence as a DEV tone in the oddball sequence, so it is not repetitive (as the STD), and therefore is free of repetition effects (e.g., repetition suppression); at the same time, it does not stand out from the statistical context (as the DEV), and therefore it is not perceived as a deviant[35, 36]. Thus, we used responses to cascades and many-standards control conditions as the reference with which to discriminate between repetition suppression and prediction error effects (Fig. 1c). If the neuronal mismatch response (DEV−STD) is caused entirely by SSA to the STD tone, responses to DEV and control conditions should remain comparable through all hierarchical levels, or if anything, the response to DEV tones should undergo a slightly stronger suppression than to the controls, due to cross-frequency adaptation[26] (Fig. 1d). By contrast, under the predictive coding framework, deviance detection is based on Bayesian inference[10], such that stronger prediction errors will be produced as more sensory input accumulates to increase the confidence and precision of current predictions[3, 12, 13]. Therefore, stronger prediction errors should be elicited by DEV than by cascades or many-standards tones, due to the lack of sequential stimulus repetitions in the controls[3, 35], and this effect should increase up the hierarchy (Fig. 1d), since higher-order processing stations are more sensitive to all forms of regularity, including complex and global regularities[2, 8, 14, 37, 38].

Our results show that the responses of lemniscal neurons were mostly dependent on tone frequency, with little sensitivity to the

different conditions. This was particularly true at subcortical levels (See Fig. 2 for individual responses of representative neurons). However, in the auditory cortex (Fig. 2, right column), strong response suppression to STD was apparent in both $AC_L$ and $AC_{NL}$, although the suppression was clearest and strongest in the non-lemniscal regions (Fig. 2b). Also, a higher firing rate in

response to DEV tones, as compared to both many-standards and cascades control conditions, was consistent across tested frequencies. These results demonstrate the hypothesized signature of prediction error at the single neuron level[15, 26].

Neuronal responses to many-standards and cascades conditions were not statistically different from each other, either in the

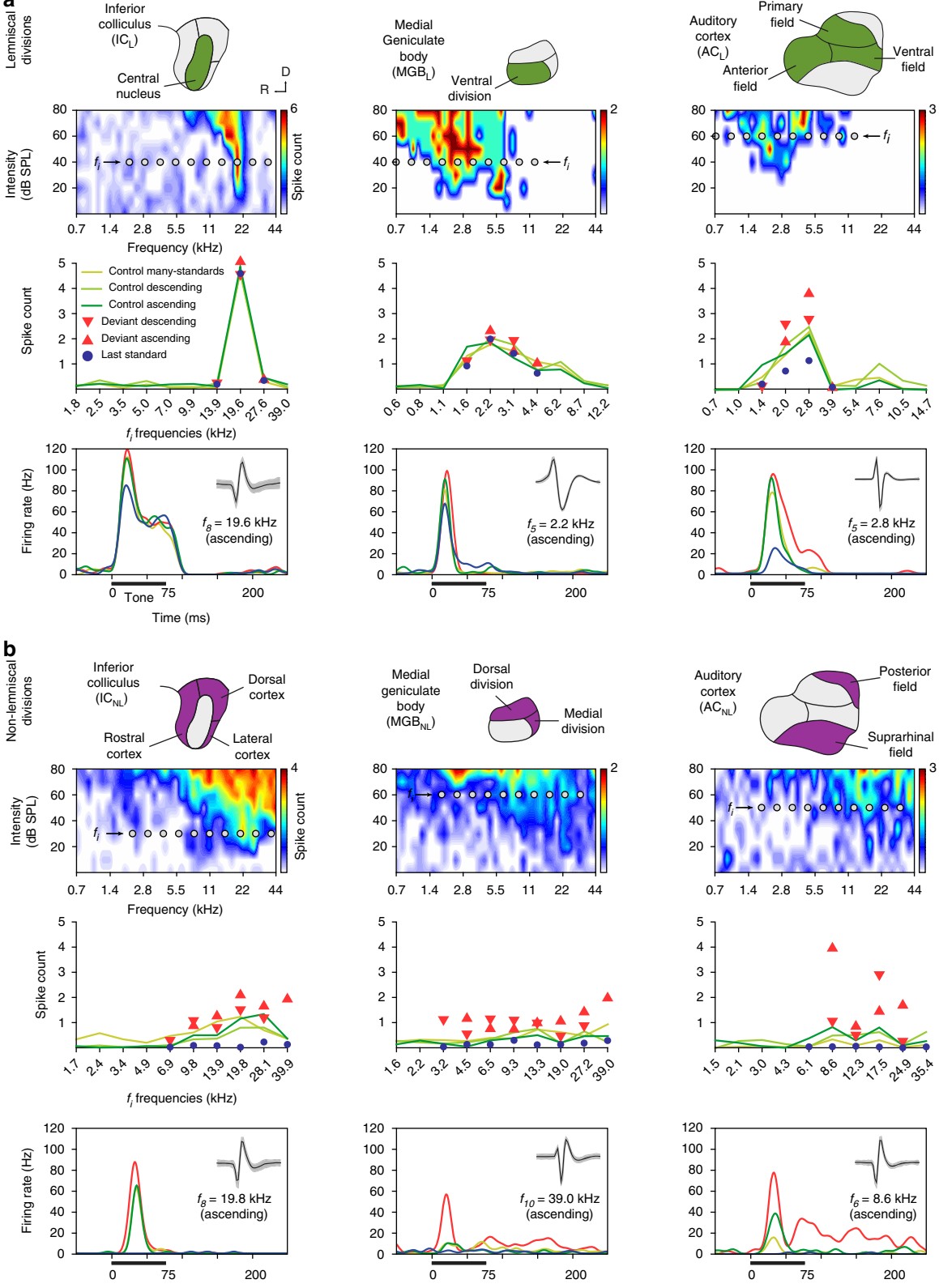

whole sample (Wilcoxon signed-rank test, $n = 1495$ $z = −0.125$, $p = 0.9$), or within each station separately (Wilcoxon signed-rank test, $n(IC_L) = 149$; $n(IC_{NL} = 522$; $n(MGB_L) = 77$; $n(MGB_{NL}) = 211$; $n(AC_L) = 250$ and $n(AC_{NL}) = 307$ $p > 0.1$ within all stations). Additionally, the results using either cascades or many-standards conditions as a control were largely comparable (Table 1). Therefore, we will limit our results to those obtained using the cascaded sequence as control (CTR), since this sequence controls for additional factors beyond presentation rate of the deviant tone[35, 36] (see Methods subsection 'Experimental design').

**Prediction error increases along the auditory hierarchy.** In order to demonstrate deviance detection at the cellular level, responses to deviant tones must exceed responses to control tones at the population level. To determine whether this was true for our data, we first performed a within-station multiple comparison (Friedman test), between responses to DEV, STD, and CTR conditions such that each pair of conditions, within each station, was tested for a difference in medians (Table 1). As expected, responses to DEV condition were stronger than to STD condition within all stations (Friedman test and median values from $IC_L = 0.12$ $p = 4.8 \times 10^{-5}$; $IC_{NL} = 0.78$ $p = 1.8 \times 10^{-92}$; $MGB_L = 0.52$ $p = 8.4 \times 10^{-6}$; $MGB_{NL} = 0.54$, $p = 7.4 \times 10^{-26}$; $AC_L = 0.71$ $p = 5.3 \times 10^{-42}$ and $AC_{NL} = 0.76$, $p = 6.7 \times 10^{-56}$; Table 1). This neuronal behavior has been described along the auditory pathway[2], but has been referred to as SSA in previous studies[19, 21, 22, 27]. However, as we demonstrate, these responses could also arise from deviance detection. Indeed, the neuronal mismatch results we showed were mostly due to the suppression of the response to the repetitive STD condition (repetition suppression), since responses to STD were significantly weaker than to CTR condition within all stations (Friedman test and median values from $IC_L = 0.41$ ($p = 2.6 \times 10^{-21}$); $IC_{NL} = 0.76$ $p = 1.6 \times 10^{-73}$; $MGB_L = 0.53$ $p = 1.3 \times 10^{-11}$; $MGB_{NL} = 0.42$ $p = 2.9 \times 10^{-16}$; $AC_L = 0.53$ $p = 4.8 \times 10^{-29}$ and $AC_{NL} = 0.38$ $p = 9.1 \times 10^{-19}$; Table 1). Critically, responses to DEV tones were already significantly higher than to CTR within the $IC_{NL}$ (median = 0.02 $p = 0.024$), and this difference increased progressively in the $MGB_{NL}$ (median = 0.12 $p = 0.019$), and $AC_{NL}$ (median = 0.38 $p = 4.9 \times 10^{12}$) (Table 1). Therefore, neuronal responses showed clear signs of prediction error at the population level, within all non-lemniscal stations, (i.e., the dorsal, lateral and rostral regions of IC, the dorsal and medial divisions of the MGB, and posterior auditory field and suprarhinal auditory field in AC) and also within $AC_L$; Table 1), which is consistent with the observed effects in the example neurons shown in Fig. 2, corresponding with $AC_L$, $IC_{NL}$, $MGB_{NL}$ and $AC_{NL}$.

To both quantify the relative contribution of repetition suppression and prediction error to neuronal mismatch in observed neuronal responses, and to facilitate comparisons between different neurons/stations, we normalized the neuronal responses to the three conditions (DEV, STD, CTR) for each neuron/tone combination. We applied Euclidean vector normalization (Supplementary Fig. 1) such that all normalized responses ranged between 0 and 1. Then, we computed three indices as the difference between normalized responses to pairs of conditions, ranging between −1 and +1 (Fig. 3a). The "index of neuronal mismatch", iMM = DEV−STD, is the relative difference in responses to STD and DEV tones in the oddball paradigm. The iMM is quantitatively equivalent to the typical SSA index[19], used in previous studies (Supplementary Fig. 2). The "index of neuronal repetition suppression", iRS = CTR − STD, is the relative reduction of the response to a standard tone, as compared to the control. Thus, the iRS quantifies repetition effects[11]. Finally, and most importantly for this study, the "index of neuronal prediction error", iPE = DEV − CTR, is the relative increase in the response to a deviant tone, compared to the control. A positive iPE reflects predictive activity[35], as opposed to SSA, and quantifies the proportion of prediction error accounting for neuronal mismatch. Therefore, the relation iMM = iRS + iPE provides a functional, quantitative decomposition of neuronal mismatch (Fig. 1d). The distribution of these indices across stations revealed that both the index of neuronal mismatch (the relative difference in responses to STD and DEV tones in the oddball paradigm) and index of prediction error (the relative increase in the response to a deviant tone, compared to the control) increase along the auditory pathway, from $IC_L$ to $AC_{NL}$ (Fig. 3b). Medians of iMM for $IC_L = 0.14$; $IC_{NL} = 0.49$; $MGB_L = 0.34$; $MGB_{NL} = 0.52$; $AC_L = 0.50$ and $AC_{NL} = 0.60$. Medians of iPE along the auditory pathway $IC_L = −0.08$; $IC_{NL} = 0.03$; $MGB_L = −0.12$; $MGB_{NL} = 0.06$; $AC_L = 0.11$ and $AC_{NL} = 0.27$.

Summary statistics for these normalized responses and indices are shown in Fig. 4a, b, respectively. Critically, median iPE was significantly greater than zero within $AC_L$ ($p = 0.01$) and within the three non-lemniscal stations (Friedman test to $IC_{NL}$ $p = 0.024$; $MGB_{NL} = 0.019$ and $AC_{NL} = 4.9 \times 10^{-12}$) (Table 1; Fig. 4b), which is consistent with a significant difference in absolute spike counts (median and p values from $IC_{NL} = 0.02$ $p = 0.024$; $MGB_{NL} = 0.12$ $p = 0.019$; $AC_L = 0.18$ $p = 0.017$ and $AC_{NL} = 0.38$ $p = 4.9 \times 10^{-12}$) (DEV−CTR in Table 1). Moreover, the iPE showed a distinct increase in two ways: (1) from lemniscal (IC = −0.08; MGB = −0.12 and AC = 0.11) to non-lemniscal stations (IC = 0.03; MGB = 0.06 and AC = 0.27); and (2) from IC to MGB to AC (Fig. 4b). To validate these observations statistically, we fitted a linear model for the iPE using "nucleus" (IC, MGB, AC) and "hierarchy" (Lemniscal "L", Non-Lemniscal "NL") as categorical factors. Using 'L' and 'IC' as reference levels for these factors, the resulting

**Fig. 2** Prediction error in representative examples of neuronal responses in anaesthetized rat. **a** Examples of lemniscal neuronal responses in each recorded auditory station (columns). The first row contains schematics of the lemniscal subdivisions (green) within each nuclei. The second row shows the frequency-response area (representation of neuronal sensitivity to different frequency-intensity combinations) of representative lemniscal neurons from each nucleus. Ten grey dots within each frequency-response area represent the ten tones ($f_i$) selected to build the experimental sequences (see Methods). The third row displays the measured responses of the particular neuron to each $f_i$ tone (baseline-corrected spike counts, averaged within 0–180 ms after tone onset) for all conditions tested. Note that measured conditions tend to overlap in the subcortical stations ($IC_L$ and $MGB_L$), and only start differentiating from each other once auditory information reaches the cortex ($AC_L$). The fourth row contains sample peri-stimulus histograms comparing the neuronal responses to each condition tested for an indicated $f_i$ tone. A thick horizontal line represents stimulus duration. A small inset within the upper right corner of each panel features the isolated spike (mean ± SEM) of that single neuron. **b** Examples of non-lemniscal neuronal responses in each recorded auditory nuclei, organized as in **a**. The first row highlights non-lemniscal divisions in purple. In the second row, note frequency-response areas tend to be more broadly tuned, as compared to lemniscal neurons. In the third row, responses to deviant conditions tend to relatively increase and distance themselves from their corresponding controls as information ascends in the auditory pathway. Also note that responses to last standards are feeble or even completely missing across all non-lemniscal stations ($IC_{NL}$, $MGB_{NL}$ and $AC_{NL}$). In the last row, the strong influence of the experimental condition over the neuronal response to the same tone can be clearly appreciated in the three nuclei

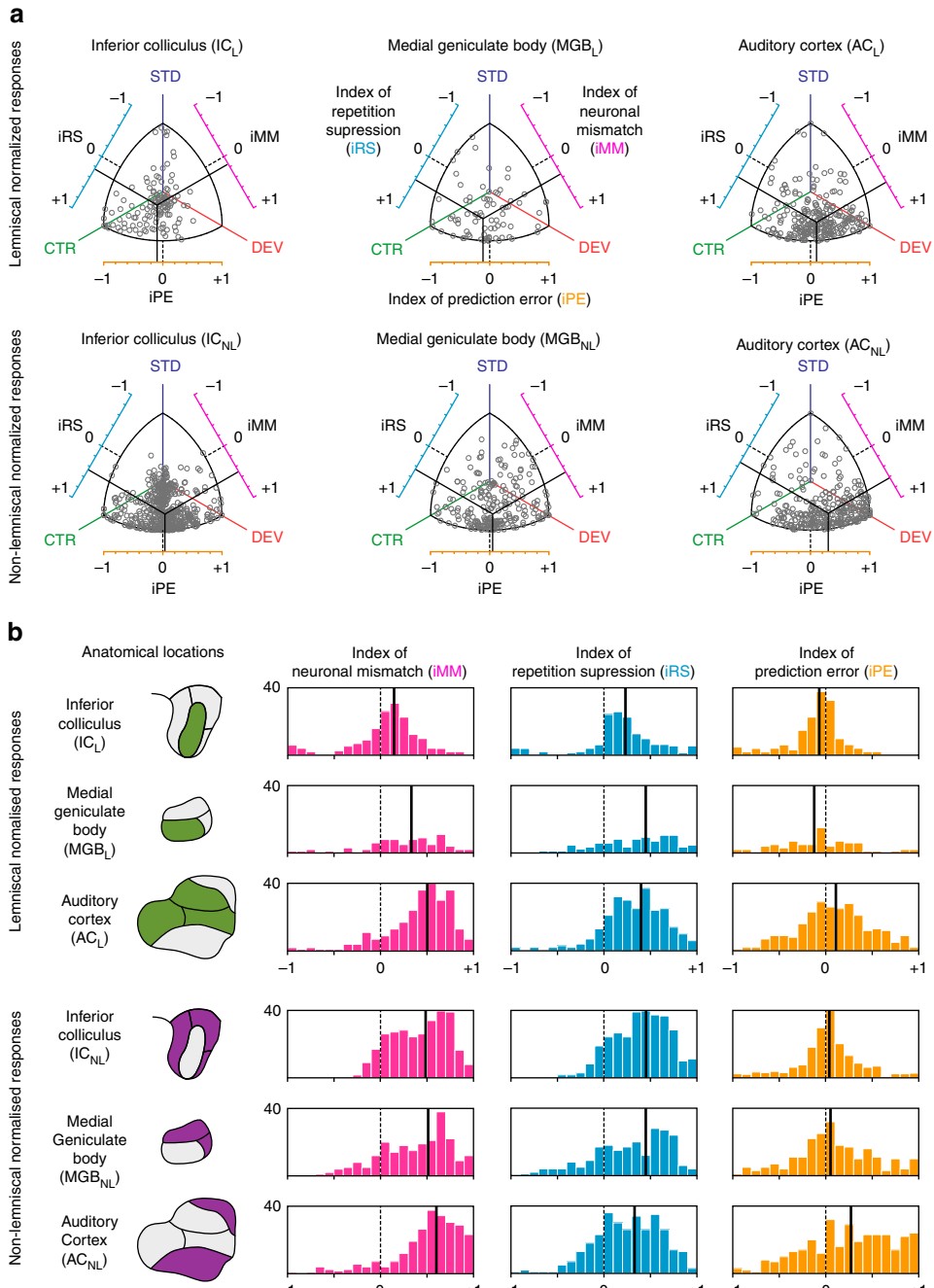

**Fig. 3** Prediction error at population level for each station in anaesthetized rat. Distribution of normalized responses and related indices of neuronal mismatch (iMM), repetition suppression (iRS) and prediction error (iPE). **a** Each grey dot in these scatter plots represents the three normalized responses of a single neuron to the same tone played as deviant (DEV), as standard (STD) and as control (CTR). Indexes result from the difference between two of these normalized responses, represented in the axes surrounding the scatter plots, where the dotted black lines marks the absence of difference between conditions (index = 0). Solid black lines represent the mean of each index, corresponding their intersection to the center of gravity of the distribution of responses in the normalized space. Note how, while the intersection for lemniscal subcortical stations (IC_L and MGB_L) is skewed towards CTR, in their non-lemniscal counterparts (IC_NL and MGB_NL) as well as all over the cortex (AC_L and AC_NL) the center of gravity of the distribution shifts closer and closer to DEV as it moves up in the auditory pathway, increasing the iPE as auditory information reaches higher-order stations. **b** Histograms represent distributions within stations of the three indexes for each neuronal response. Solid black lines indicate medians. The noticeable overall tendency of the median indexes to shift towards more positive values, from IC through MGB to AC, and from lemniscal to non-lemniscal divisions, unveils a hierarchy of processing in the auditory pathway

model was:

$$iPE = 0.012 + 0.020 \times NL - 0.136 \times MGB + 0.092 \times AC$$
$$+ 0.185 \times NL \times MGB + 0.158 \times NL \times AC$$

where the constant term 0.012 is the reference level in IC_L. Then, we applied an ANOVA to this model and revealed a significant effect of hierarchy ($F = 36.43$, $p = 2.01 \times 10^{-9}$) and nucleus ($F = 45.74$, $p = 5.53 \times 10^{-20}$), and a significant hierarchy×nucleus

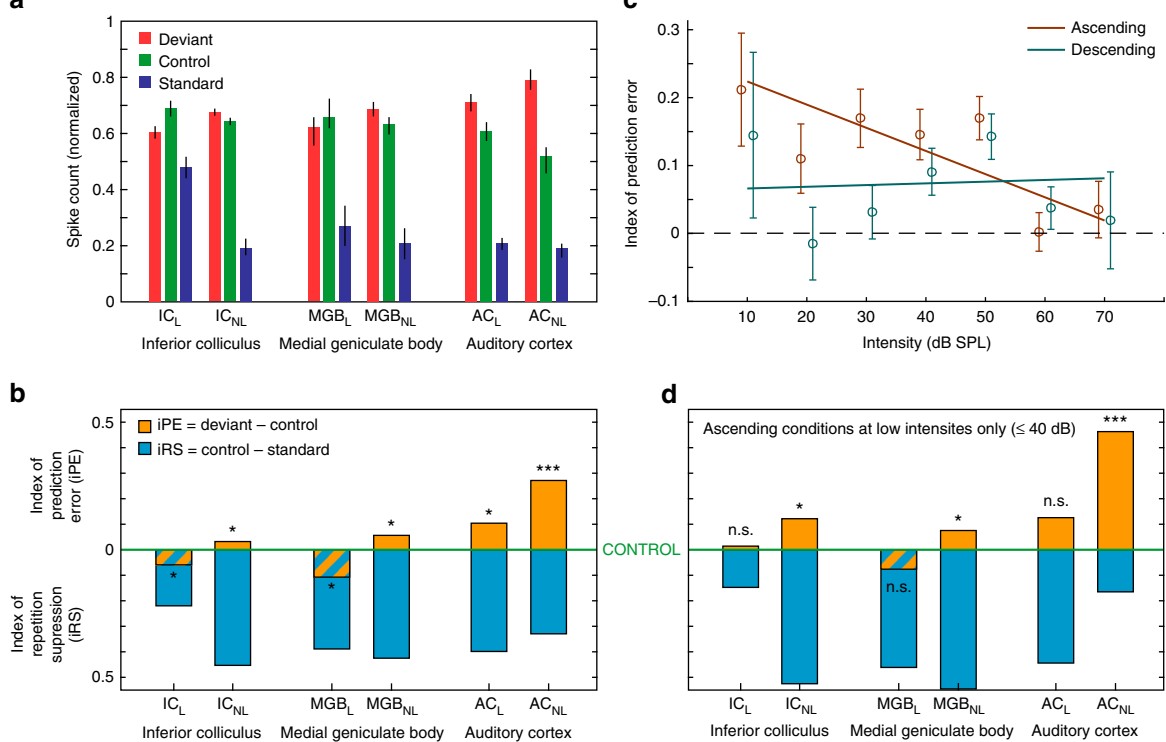

**Fig. 4** Emergence of prediction error along the auditory hierarchy. **a** Median normalized responses (lines indicate SEM) to the deviant, standard and control within each station. **b** Median indices of prediction error (orange) and repetition suppression (cyan), represented with respect to the baseline set by the control. Thereby, iPE is upwards-positive while iRS is downwards-positive. Each median index corresponds to differences between normalized responses in **a**. Asterisks denote statistical significance of iPE against zero median (*p = 0.05, **p = 0.01, ***p = 0.001, see Table 1). **c** Linear model fitted for the iPE using SPL and direction (ascending / descending) as predictors. Error bars denote mean and SEM for each SPL and direction. Note the model predicts greater iPE values for ascending conditions at low intensities. **d** Same as in **b**, but only representing ascending conditions at intensities equal or lower than 40 dB SPL

interaction ($F = 3.7$, $p = 0.024$). Therefore, both tendencies, from lemniscal to non-lemniscal and from IC to MGB to AC, were significant and robust from midbrain to cortex. Specific post hoc comparisons confirmed that median iPE was higher in $AC_{NL}$ than in $AC_L$ ($n = 557$ ranksum test, $p = 2.2 \times 10^{-5}$) or $MGB_{NL}$ ($n = 518$ $p = 1.9 \times 10^{-5}$), and higher in $AC_L$ than in $IC_{NL}$ ($n = 773$, $p = 2.2 \times 10^{-13}$). Although iPE was numerically higher in $MGB_{NL}$ than in $IC_{NL}$, this difference was not quite statistically significant ($n = 734$, ranksum test, $p = 0.151$).

Overall, this analysis demonstrates a systematic increase of prediction error in responses of single neurons as information progresses along the auditory pathway. This was true, both from the IC to the MGB to cortex (bottom-up processing) and from lemniscal to non-lemniscal regions, with a mutual potentiation of these two effects.

According to previous modeling work, single neurons were expected to be maximally sensitive to change for stimulus ranges, where the firing rate of the neuron is below saturation[39]. Consistent with this hypothesis, we observed that deviance specific-responses were easier to produce with low stimulation intensities, particularly for ascending deviants (e.g., Fig. 2d, $IC_{NL}$). To test these observations at the population level, we fitted a different model for the iPE, using SPL (in Bels = dB SPL/10) and direction (ascending or descending) of deviant tones (see Fig. 1b) as predictors. The model showed a significant effect of SPL ($F = 4.59$, $p = 0.03$) and a SPL×direction interaction ($F = 6.66$, $p = 0.01$):

$$\text{iPE} = 0.064 + 0.194 \times \text{ascending} + 0.003 \times \text{SPL}$$

$$- 0.037 \times \text{ascending} \times \text{SPL}$$

which indicates that the iPE is expected to be much higher for ascending deviants at intensities equal or below 40 dB SPL (Fig. 4c). Indeed, we observed a distinct increase in the iPE within all stations (medians and Friedman test from $IC_L$ $n = 15$; median = −0.003 and $p = 1$; $IC_{NL}$ $n = 113$; median = 0.1174 and $p = 0.0052$; $MGB_L$ $n = 12$; median = −0.0739 and $p = 0.6831$; $MGB_{NL}$ $n = 40$; median = 0.1041 and $p = 0.0442$ and $AC_L$ $n = 61$; median = 0.1364 and $p = 0.0629$), under these stimulation conditions (Fig. 4d), particularly in $AC_{NL}$ ($n = 38$ median = 0.5048 and $p = 1.01 \times 10^{-4}$), where prediction error accounted for around two thirds of the iMM.

We also wanted to test the relationship between a neuron's deviance sensitivity and its tuning width, since broadly tuned neurons have wider spectral integration capabilities, which in turn might facilitate the task of deviance detection. Specifically, broadly tuned neurons would be activated by more of the control tones than narrowly tuned neurons, which in turn could reduce neuronal responses to the control condition in broadly tuned neurons (compared to narrowly tuned neurons, which are more abundant in lemniscal stations). However, we did not find any significant correlation between neuronal tuning bandwidth (measured as a quality factor $Q_{30}$, see Methods section) and iPE in our sample (Spearman correlation coefficient, $\rho = -0.0067$, $p = 0.93$). Interestingly, however, a subset of neurons with highly disorganized and fragmented frequency-response area, for which a $Q_{30}$ factor could not be measured, showed iPE levels significantly higher than the rest (median iPE, ranksum test; untuned neurons: iPE = 0.31, tuned neurons: iPE = 0.024, $p = 1.6 \times 10^{-5}$). This indicates that the functional role of these neurons are more concerned with contextual integration at a higher level than with spectral processing.

**Single neuron PE and large-scale mismatch response in AC.** We used the same electrodes from which we recorded the single neuron spike to simultaneously record local field potentials (LFPs). We then leveraged these latter signals to explore the direct correlation between the prediction error demonstrated in the spike responses and large-scale mismatch responses (such as the MMN). We averaged LFP responses for each condition and station, as well as the difference between DEV and CTR conditions, which we called the "prediction error potential"[16, 36]: PE-LFP = $\text{LFP}_{DEV} - \text{LFP}_{CTR}$ (Fig. 5). A significant early PE-LFP using a two-

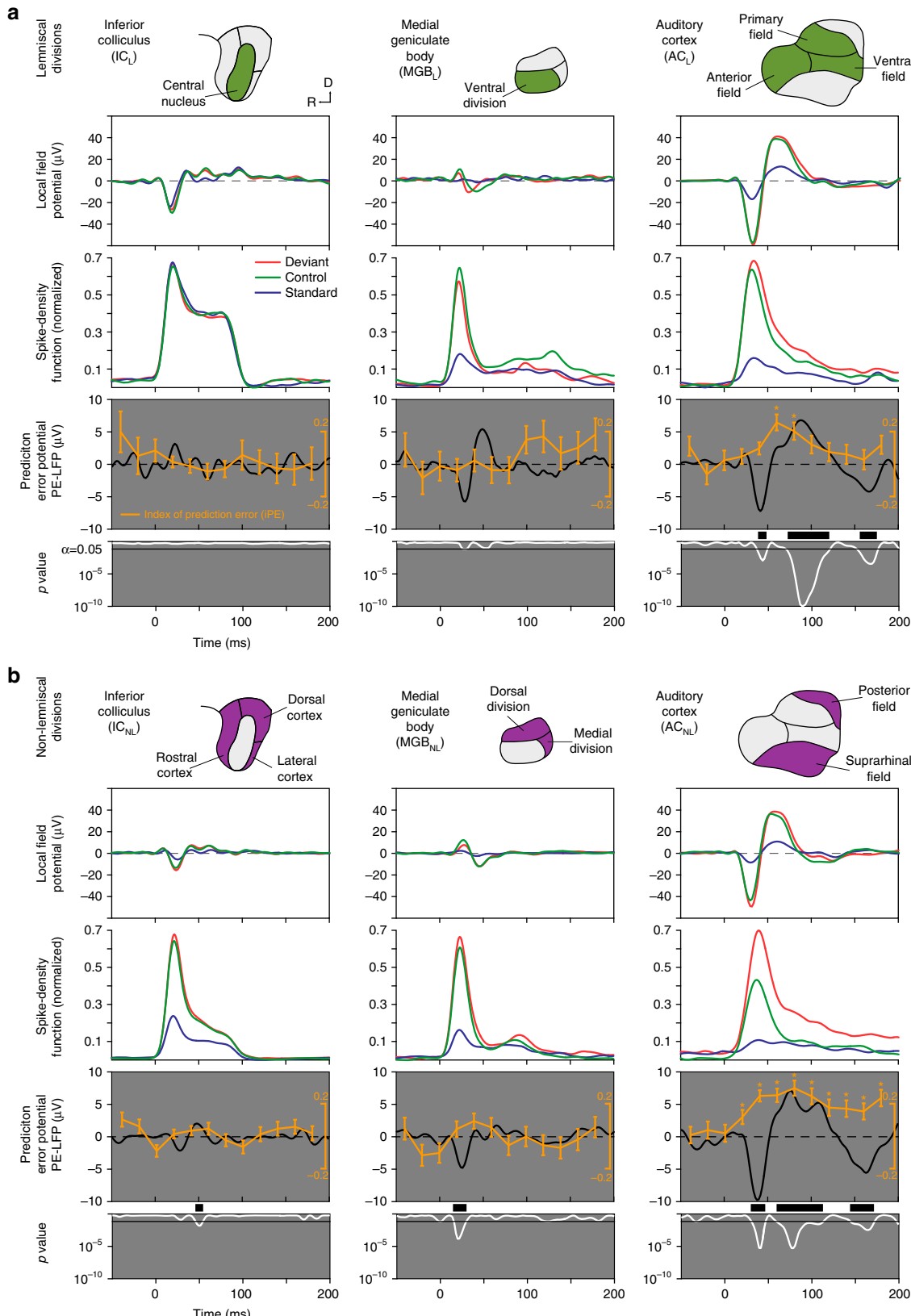

tailed $t$ test was already detectable within $IC_{NL}$ (median = 0.0480 and $p < 0.05$) and $MGB_{NL}$ (median = 0.0550 and $p < 0.05$) (Fig. 5b, left and central columns). In the auditory cortex, the PE-LFP was strong and significant in both $AC_L$ and $AC_{NL}$ (median = 0.1269 and 0.4929 respectively and $p < 0.05$), showing three major deflections (Fig. 5, right column): a fast negative deflection (N1; 35–50 ms after change onset), a slower positive deflection (P2; 70–120 ms), and a third, late, negative deflection (N2; beyond 150 ms; paired $t$ test, FDR-corrected for 200 comparisons). Epidural MMN peaks between 60 and 120 ms in rats[15], the same range of the P2 recorded here for the PE-LFP, and can be positive when recorded from inside the brain[29]. Then, we recomputed the iPE for 12 different time windows (20 ms width, from −50 to 190 ms respect to stimulus onset), for each neuron/tone combination separately, and we averaged within each station (Fig. 5). The iPE showed a clear modulation over time in both $AC_L$ and $AC_{NL}$ stations (Friedman test, not corrected for 6 independent tests). Each individual iPE value was also tested against zero (signrank test, FDR-corrected for 12 comparisons), and this analysis revealed a significant iPE ($p < 0.05$ in both asterisks) within $AC_L$ between 60–100 ms after change onset, and in $AC_{NL}$ ($p < 0.01$ for all asterisks) between 25–200 ms, and seemingly beyond (Fig. 5, right column). In summary, the highest iPE values, which reflect prediction error in single neuron responses, correlate in time and location with a large-scale mismatch wave (the PE-LFP), which is the putative MMN in rats[15, 16].

**Hierarchical prediction is conserved across species and arousal**. The pattern of results shown in Fig. 4b suggests that the auditory system adheres to the general predictive coding framework. To confirm both that these results held different species of rodent, and to exclude the potential biasing influence of anesthesia, we performed identical experiments in awake, restrained mice. We recorded multi-unit activity and LFP from IC (49 recordings from 5 animals) and AC (42 recordings from 5 animals). Representative sample recordings are shown in Fig. 6. As in single-unit cases recorded in the anesthetized preparation (Fig. 2), we observed clear signs of prediction error (DEV > CTR) across different frequencies in the multi-unit recordings (Fig. 6).

Normalized responses and indices of prediction error and repetition suppression for the awake mice are shown in Fig. 7 and Table 2. Similarly to the results obtained from the anesthetized rats, median iPE was significantly greater than zero within $IC_{NL}$ and both AC stations (median iPE, Friedman test with post hoc multiple comparisons, Fisher's Least Significant Difference method; $IC_L$: −0.13, $p = 0.023$; $IC_{NL}$: 0.12, $p = 0.018$; $AC_L$: 0.17, $p = 0.001$; $AC_{NL}$: 0.32, $p = 5 \times 10^{-7}$). Also, iPE was significantly higher in $AC_{NL}$ than in $AC_L$ (ranksum test, $p = 0.015$), or in $IC_{NL}$ ($p = 0.0004$). Therefore, the two extremes of the hierarchical organization of the iPE (IC and AC) coincide in awake mice and anesthetized rats (compare Figs. 4a, b and 7b, c). This finding is consistent with the hypothesis that neurons along the auditory pathway exhibit a hierarchical organization of prediction error is a general pattern across rodent species and states of awareness. Indeed, median iPE levels in both $AC_L$ and $AC_{NL}$ were not statistically different between the two preparations (ranksum test, $p > 0.1$). However, median iPE levels were significantly higher in awake than in anesthetized $IC_{NL}$ (ranksum test, $p = 0.048$; compare above values with Table 1). Thus, iPE tended to be higher in the awake condition, within each processing station (compare Figs. 4a, b and 7b, c), especially in subcortical $IC_{NL}$. Finally, as was the case in the anesthetized rat, a difference between DEV and CTR conditions was also observed in the LFP, at the level of the AC ($t$ test $p < 0.05$ for lemniscal and non-lemniscal regions. Fig. 7a, third row); and the difference was significant during similar time windows: 35–42 ms (N1) and 95–118 ms (P2) in $AC_L$, 86–116 ms (P2) and beyond 165 ms (N2) in $AC_{NL}$ (compare with Fig. 5, right column).

## Discussion

This study demonstrates that predictive activity of single neurons responding to an auditory oddball paradigm can be tracked along the ascending auditory pathway. These prediction error signals are organized hierarchically and are consistent across species and awareness states. Furthermore, our data suggests that this predictive activity underlies large-scale mismatch responses, such as the MMN. Quantitatively decomposing neuronal mismatch responses into repetition suppression and prediction error revealed a systematic increase in the proportion of prediction error that explained the neuronal mismatch responses as the sensory signal traveled along the ascending auditory pathway. The increase in explanatory power of the prediction error signal occurred not only from the inferior colliculus to auditory thalamus and cortex, but also from lemniscal (first order) to non-lemniscal (high order) divisions within each level. Thus, the highest prediction error values are found in the higher-order auditory cortex, where they correlate with a large-scale prediction error potential including late evoked potentials.

This latter finding suggests an influence from prefrontal cortices[40]. This view is consistent with a recent study that recorded from humans subdural electrocorticographic electrodes located in frontal and temporal cortex while they listened to trains of repeated tones that were interrupted by two types of deviant: predictable and unpredictable[41, 42]. Using high gamma (Hγ-band, > 60 Hz) activity as an index of local spiking these authors found more evidence for a hierarchical organization of mismatch signals[42], highlighting the role of frontal cortex and Hγ-band activity in deviance detection and in the generation of predictive activity. Interestingly, a recent study using LFP recordings in the parietal and frontal cortex in rats also supports this notion[40]. Our finding that prediction error contributes to neuronal mismatch response supersedes repetition suppression within the higher-order auditory cortex is also consistent with studies of the neuroanatomical location of the MMN in animals[16] and humans[37].

**Fig. 5** Correlation of iPE and prediction error potential (PE-LFP). **a** Population grand-averages for different response measures, computed for each lemniscal station (in columns, represented in first row highlighted in green). The second row shows the average LFP across all tested tones and single neurons from each station for different conditions. The third row displays the average firing rate profiles for each station as normalized spike-density functions. The fourth row contains the prediction error potentials (PE-LFP, black trace), which is the difference wave of the deviant and the control LFP. Along PE-LFP, the time course of the average iPE is plotted in orange (mean ± SEM, asterisks indicating significant iPE for the corresponding time window; Wilcoxon signed-rank test for 12 comparisons, corrected for FDR = 0.1). Next row shows an instantaneous $p$ value (white trace) of the corresponding PE-LFP (paired $t$ test against equal means, corrected for FDR = 0.1, critical threshold for significance set at 0.05 represented as a horizontal bar). Thick black bars of the grey panel mark time intervals for which the average PE-LFP is significant. Note that only $AC_L$ shows a significant prediction error signal. **b** Same as in **a** but computed for each non-lemniscal station (highlighted in purple in the first row). Note in the last row significant PE-LFPs appear in all three stations ($IC_{NL}$, $MGB_{NL}$ and $AC_{NL}$), and prominently in $AC_{NL}$. Note also how highest iPE values are concurrent with the strongest PE-LFPs in time and location (auditory cortex, both $AC_L$ and $AC_{NL}$)

Taken together, results from previous studies cohere with our findings and present strong evidence for the predictive coding account of mismatch responses. Our study also extends this work, highlighting the role of subcortical structures in perception[43], providing a novel extension of the exclusively cortical perspective of the predictive coding literature[9, 10, 44]. Although lemniscal and non-lemniscal pathways process different aspects of the auditory signal in parallel, the non-lemniscal auditory regions represent a higher hierarchical level of processing[33] and are known to be more sensitive to acoustic change and contextual influences than lemniscal ones[2, 22, 25, 27, 45]. In fact, the involvement of higher-order areas in predictive processes has been hypothesized

previously[4], but until now, this hypothesis had not been tested directly.

The response patterns we observed confirm that subcortical, first-order nuclei are mostly sensitive to global or pattern probability generated in the classical oddball paradigm, while higher levels are more sensitive to local relationships between sounds (transitional probabilities), exactly as observed in human MMN studies[42,46]. Thus, our data are consistent with a passive stimulus-specific adaptation underlying oddball responses in the lemniscal midbrain and thalamus[26,47]. By contrast, the responses we observe in high-order regions support a generative mechanism of Bayesian inference being at play in auditory cortex and high-order subcortical stations of perceptual processing[3]. The contrast between first-order and high-order neuronal mismatch is particularly clear within the auditory thalamus. Responses to the deviant condition are more adapted than to the cascade sequence condition, exactly as predicted by the SSA model in narrow frequency channels[26]. However, median of the index of neuronal prediction error is significantly positive at the high-order thalamus, indicating actual prediction error. Thus, in the case of prediction error, we have shown that the higher-order midbrain and thalamus behave like the auditory cortex. It is likely that the enhancement of responses to deviant tones seen at subcortical levels is modulated, at least in part, by top–down cortical influences[48–51], and this is precisely what the hierarchical predictive coding framework would suggest[12, 51]. Indeed, the lower levels of prediction error seen in the high-order midbrain in the anesthetized preparation, as compared to the awake condition, suggests that descending connections play a role deviance detection, and are therefore reduced by anesthesia.

The enhanced prediction error for low intensities of stimulation could facilitate perception under challenging sensory conditions, by increasing the gain of prediction error responses at early processing stages[12]. These findings parallel previous observations of single neurons of the primary visual cortex[52]. The former study showed that cortical feedback improves figure-background discrimination of low-salience stimuli[52]. The dependence of prediction error on intensity conforms with previous studies showing a bias to deviance detection being stronger at the high-frequency edges of the frequency-response areas in collicular neurons[22]. Finally, asymmetries in the direction of frequency-change detection (ascending vs. descending) have also been found in both animal[36] and human[53] MMN studies, although this asymmetry was only weak for frequency modulation tones similar to our cascaded condition[54]. Moreover, as discussed elsewhere[36] the asymmetry with respect to the direction of the deviant indicates an overall trend towards a higher sensitivity of the rodents brain to increments in frequency. The auditory system of the rodents may therefore be primed to perceive high-frequency noises like the ultrasonic vocalizations that these species use to communicate with each other[36].

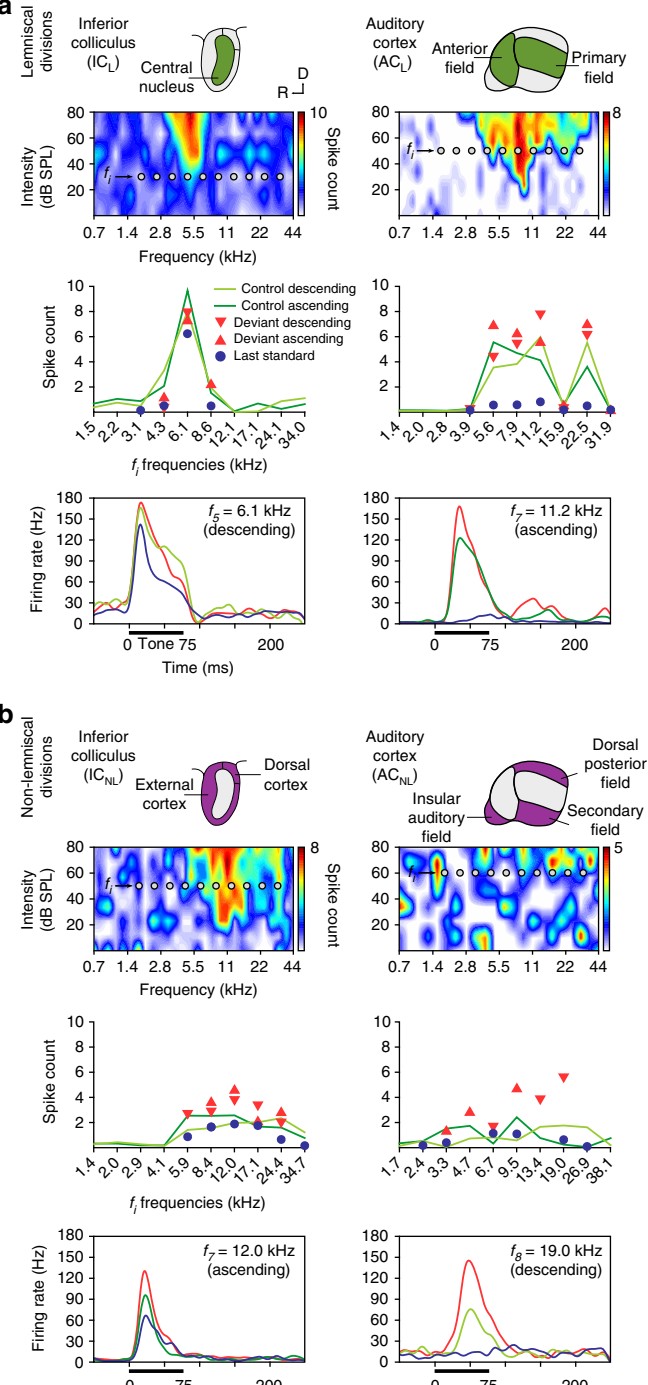

**Fig. 6** Prediction error in representative examples of neuronal responses in awake mouse. **a** Examples of lemniscal multiunit activity recorded in two auditory nuclei (columns). The first row contains schematics of the lemniscal subdivisions (green) within each station. The second row shows a frequency-response area of each nuclei. Ten grey dots within those frequency-response area represent the ten tones ($f_i$) selected to build the experimental sequences (see Methods). The third row displays the measured responses to each fi tone (baseline-corrected spike counts, averaged within 0–180 ms after tone onset) for all conditions tested. The fourth row contains sample peri-stimulus histograms comparing the neuronal responses to each condition tested for an indicated $f_i$ tone. Stimulus duration is represented by a thick horizontal line. **b** Examples of multiunit activity recorded in non-lemniscal divisions (first row, colored purple) of each auditory nuclei, organized as in **a**

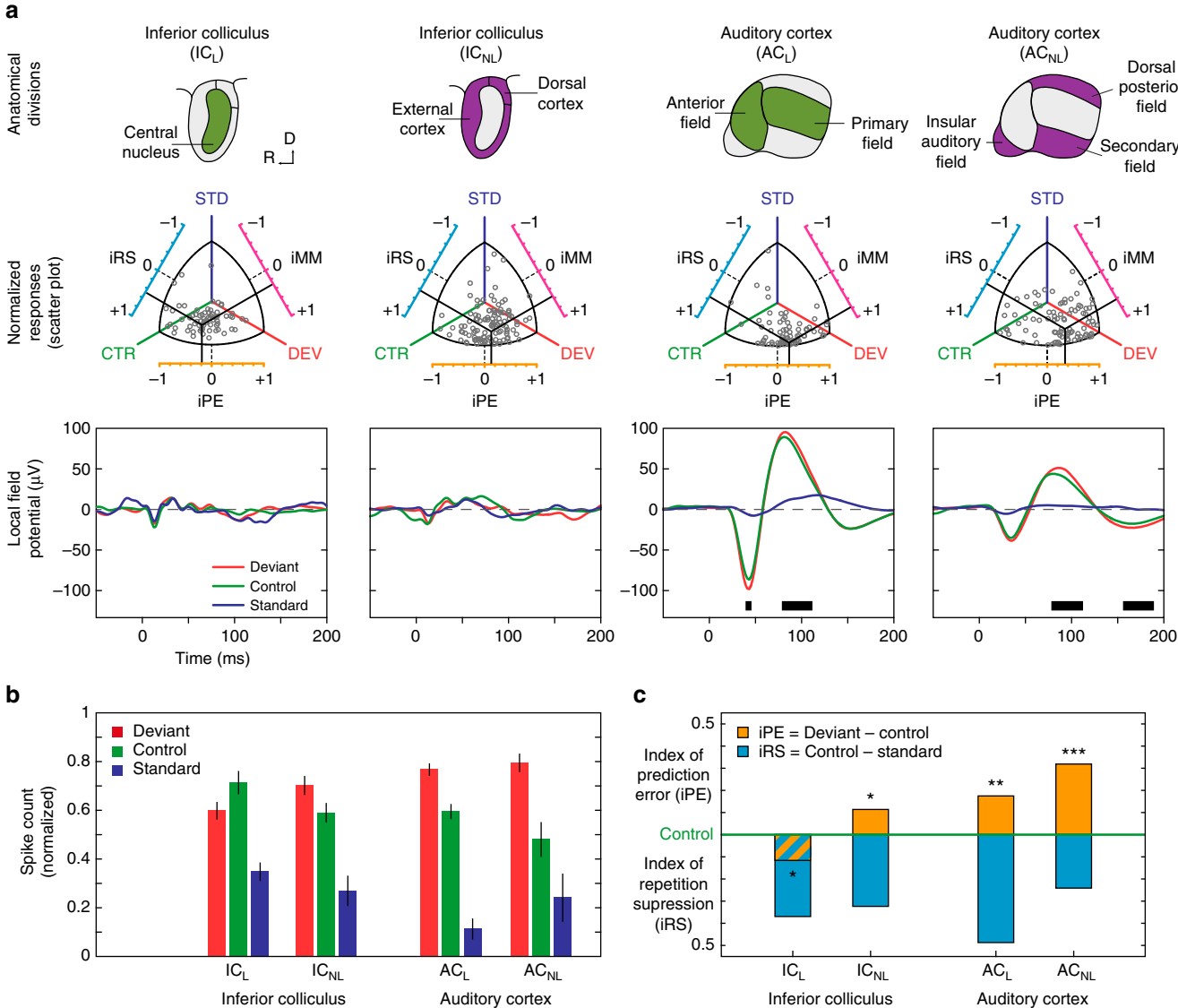

**Fig. 7** Population results for awake mouse. **a** Summary of population results for each recorded station (columns). The first row displays lemniscal (green) and non-lemniscal (purple) subdivisions of two recorded auditory nuclei of the mouse brain. The second row contains scatter plots featuring normalized responses of each multiunit recording to the same tone played as DEV, STD and CTR (grey dots) and the mean population values of each index (solid black bars). The third row contains the average LFP across all tested tones from each station for different conditions. Thick black bars at the bottom of the panels mark the time intervals were the difference between the deviant LFP and the control LFP is significant, thereby producing a prediction error potential. **b** Median normalized responses (bar indicate interquartile range) to the deviant, standard and control within each station. **c** Median indices of prediction error (orange) and repetition suppression (cyan), represented with respect to the baseline set by the control. Asterisks denote statistical significance of iPE against zero median (*$p = 0.05$, **$p = 0.01$, ***$p = 0.001$). Note the overall similarities with results in the anaesthetized rat (Figs. 3–5), confirming a hierarchical generation of prediction error also in awake preparations

Previous studies of deviance detection in rat auditory neurons were limited to primary auditory cortex, and yielded inconclusive results. In most of these studies, responses to deviant tones were not different from control tones, pointing to a purely-SSA explanation of oddball responses[26, 28], although this result was interpreted by some of these authors[26] as indicative of deviance detection, based on theoretical considerations. In this context, it is worth noting that in some experiments in rats, anesthesia with ketamine (an NMDA-antagonist) has shown a weakened MMN[55] and abolition of global mismatch responses[56]. This pattern of effects has been called a disruption of predictive coding[57]. Indeed, we observed that prediction error tended to be higher in the awake condition and this might be one important reason why deviance detection was not apparent in previous rat studies[26, 28].

A recent study in mouse primary auditory cortex has unambiguously demonstrated signs of deviance detection in late responses of single units, using the many-standards control sequence[30]. The cascade sequence is arguably a better control for repetition effects than the many-standards sequence[35]. This is so because the many-standards sequence overestimates the true state of refractoriness in the oddball whereas the cascading control is highly comparable to the deviant without violating any regularity[35]. However, so far the cascade sequence has been used in only a single animal study that yielded inconclusive results[36]. Several reasons may explain the ambiguous results. First, the use of the cascade control sequence may result in an underestimation of deviance detection, with the stimuli used as deviants are sitting at the outer extremes of the range of stimuli[36]. A second reason is

**Table 2 Summary of awake data set**

|  | $IC_L$ | $IC_{NL}$ | $AC_L$ | $AC_{NL}$ |
|---|---|---|---|---|
| Neurons | 20 | 27 | 16 | 23 |
| Points | 61 | 104 | 75 | 77 |
| DEV (spikes) | 1.3481 | 1.5188 | 3.0306 | 0.7589 |
| STD (spikes) | 0.8515 | 0.5961 | 0.4807 | 0.2141 |
| CAS (spikes) | 1.8504 | 1.4219 | 2.0772 | 0.4999 |
| DEV−STD (spikes) | 0.4966 | 0.9227 | 2.5499 | 0.5448 |
| $p$ value | 0.000 | 0.000 | 0.000 | 0.000 |
| CTR−STD (spikes) | 0.9989 | 0.8258 | 1.5965 | 0.2858 |
| $p$ value | 0.000 | 0.000 | 0.0015 | 0.0048 |
| DEV−CTR (spikes) | **−0.5023** | **0.09669** | **1.5965** | **0.2590** |
| $p$ value | 0.024 | 0.018 | 0.000 | 0.000 |
| iMM | 0.2387 | 0.4292 | 0.6612 | 0.5773 |
| $p$ value | 0.000 | 0.000 | 0.000 | 0.000 |
| iRS | 0.3663 | 0.3048 | 0.4910 | 0.2552 |
| $p$ value | 0.000 | 0.000 | 0.000 | 0.0048 |
| iPE | **−0.1276** | **0.1244** | **0.1702** | **0.3222** |
| $p$ value | 0.024 | 0.018 | 0.001 | 0.000 |

For IC and AC stations: Number of multi-unit activity recorded and tested neuron/tone combinations (points). Median values for baseline-corrected spike counts to the different conditions. Median differences between the former measures, and associated $p$ values against zero (Friedman test with post-hoc multiple comparison, Fisher's Least Significant Difference method, uncorrected for 6 independent tests). All $p$ values are rounded to 3 decimal figures, so a value of 0.000 means "$p < 0.0005$". Median indices of neuronal mismatch (iMM), repetition suppression (iRS) and prediction error (iPE), computed using the cascade control sequence, and their corresponding $p$ values (note that $p$ values are the same for absolute differences and normalized indices, since these indices are median differences between normalized responses, and the non-parametric test is independent of scaling). Values related to predictive neuronal activity are highlighted in bold case, since they represent the most significant result of this research

that the cascade control required the use of a higher frequency deviant for the ascending oddball condition. Finally, it could be arguable that the pattern of regularity established by the cascade sequence could be modeled by the rat brain[36]. Our results, using single-unit recordings, were comparable or even more robust for the cascade than for the many-standard control, in agreement with human studies[35]. Thus, although the rat brain may not be capable of fully encoding the complex regularity of the cascade control condition[36], this regularity may serve to boost the index of neuronal prediction error levels at subcortical structures just enough to make them detectable and statistically significant. This observation suggests that future research of subcortical deviance detection should use the cascade sequence as a control.

A fundamental theme in cognitive neuroscience is the generalization of predictive coding across sensory modalities and animal models. Importantly, predictive activity using a design similar to ours has been shown in sensory modalities other than audition, such as in rat barrel cortex[58], mouse visual cortex[59] and both primary and non-primary rat visual cortex[60]. The latter study found clear signs of deviance detection in latero-intermediate area in extrastriate cortex, a higher-order visual area, but only SSA in the primary visual cortex, demonstrating also a hierarchical organization, i.e., neural responses along the rat ventral visual stream become increasingly sensitive to changes in the visual environment. Although the visual system does not have the lemniscal/non-lemniscal organization[32, 33] of the auditory and somatosensory systems, recent reports have demonstrated distinct adaptation effects cascading through the visual system[60, 61]. This suggests that our results generalize across the senses and types of organization. Moreover, a mouse model of visual MMN found that both MMN and schizophrenia are based on the same underlying sensory deficits[59]. Despite these similarities, caution should be taken when equating sensory modalities

between species[59]. Indeed, our results also contrast with previous studies that show little or no evidence of predictive coding in the auditory cortex of monkeys and humans[28, 29, 42, 62]. Anatomical/functional and/or methodological differences likely account for some of the discrepant findings. Rodents and primates have different auditory anatomical/functional organizations. These differences are most apparent and pronounced at the cortical level[34] such that more complex or sophisticated functions may occur at lower levels of the system in rodents[63]. Specifically, the complex computational machinery of the subcortical auditory system led some authors to speculate a comparable computational role of the inferior colliculus and the primary visual cortex[63]. Technical differences may also account for the discrepancies with our current results. While we used mostly single-unit recordings, previous studies carried out in monkeys and humans[29, 62] and even previous rodent studies[26, 28] used local field potentials, current source density components, multiunit activity, and/or Hγ-band responses. These techniques are excellent for population activity, but they measure aggregate local synaptic input rather than neuronal output and do not pick up activity patterns that are present at a finer neuronal level.

Our study suffers from some technical limitations as well. While we made electrolytic lesions in IC and MGB consistently, we did not mark recording sites in AC and therefore our results are inconclusive about the layer organization of our AC recordings. According to the canonical circuit of predictive coding, error units and prediction units are differentially located in supra-granular and infragranular layers, respectively[9, 41, 64]. Future studies using, for example, patch-clamp recording to label the individual neurons (including their axonal arborizations) could address this issue and would help to disentangle the differences between local feedforward and feedback processing within and across layers. Another important caveat to our study is that we do not investigate the relationship between prediction and attention. Although this question was outside the scope of the study, it is worth mentioning that predictive coding is associated with different cortical rhythms[9, 41]. Error units seem to propagate messages forward via gamma-band (high frequency) while prediction units propagate via lower beta-band (low frequency)[64]. The selective Hγ-band amplitude modulation to unpredictable deviants mentioned above might also reflect a switch of attention[42]. Future experiments using recordings in animals to study cortical rhythms and frontal cortex responses might provide a more detailed and refined picture for the relation between predictive coding and attention.

In conclusion, our results demonstrate that prediction error is a fundamental component of responses of single auditory neurons to an auditory oddball paradigm. This prediction-error signal is detectable even at subcortical levels, thereby adding additional evidence in support of the predictive coding framework of perceptual processing. In addition, we show that neuronal predictive activity underlies the generation of large-scale mismatch responses in animal models, paralleling fundamental properties of the human MMN such as the hierarchically organization of prediction error along the central auditory pathway. Critically, we have shown that our results hold across rodent species and arousal and hence, we have validated rodent preparations as animal models of MMN. These are promising results for translational research into the cellular mechanisms of neural disorders characterized by reductions in large-scale mismatch responses, such as the MMN.

## Methods

**Experimental design.** Experiments in anesthetized rats were performed on 36 adult, female Long-Evans rats with body weights between 200–250 g (aged 9 to 15 weeks). The experimental protocols were approved by, and used methods

conforming to the standards of, the University of Salamanca Animal Care Committee and the European Union (Directive 2010/63/EU) for the use of animals in neuroscience research.

Sounds used for stimulation were white noise bursts or pure tones with 5 ms rise-fall ramps. Sounds used for searching for neuronal activity were trains of noise bursts or pure tones (1–8 stimulus per second). We used short stimulus duration for searching (30 ms) to prevent strong adaptation. In addition, type (white noise, narrowband noise, pure tone) and parameters (frequency, intensity, presentation rate) of the search stimuli were varied manually when necessary to facilitate release from adaptation, and thus prevent overlooking responses with high SSA. All stimuli presented were sinusoidal pure tones of 75 ms duration, including 5 ms raise/fall ramps.

For each recorded neuron, the frequency-response area that is the map of response magnitude for each frequency/intensity combination was first computed (Fig. 2, second and sixth rows). To obtain this frequency-response area, a randomized sequence of tones was presented at a 4 Hz rate, randomly varying frequency and intensity of the presented tones (3–5 repetitions of all tones). Then, we selected 10 evenly-spaced tones (0.5 octave separation) at a fixed sound intensity (usually 20–30 dB above minimal response threshold), so that at least two of them fell within the frequency-response area or close to its limits (Figs. 1b and 2). These 10 frequencies were used to create the control sequences shown in Fig. 1c. Additionally, adjacent pairs of them were used to present different oddball sequences. All sequences were 400 tones in length, at the same, constant presentation rate of 3 Hz (for AC) or 4 Hz (for IC and MGB). A faster presentation rate was used for subcortical recordings, to compensate for the relative slowing down of preferred repetition rates from brainstem to cortex[34].

We used oddball sequences[5, 19] (Fig. 1b) to test the specific contribution of deviance to the neuronal responses. An oddball sequence consisted of a repetitive tone (standard 90% probability), occasionally replaced by a different tone (deviant 10% probability), in a pseudorandom manner. The first 10 tones of the sequence were always the standard tone, and a minimum of 3 standard tones always preceded each deviant. Oddball sequences were either ascending or descending, depending on whether the deviant was a higher or lower frequency than the standard (Fig. 1b). To control for the overall presentation rate of the target tone, we used two different control sequences, namely, the many-standards and cascaded sequences[26, 35] (Fig. 1b). The many-standards control sequence was a random presentation of the 10 selected tones, such that each of them was played the same number of times in an unpredictable order but a single tone was never repeated. Two cascaded control sequences, ascending and descending, contained the same 10 tones but were arranged according to ascending/descending frequency, respectively (Fig. 1b). Since all sequences were 400 stimuli long, a tone was played with the same overall presentation rate (4 Hz) in the deviant, many-standard control sequence and cascade control sequence conditions, a total of 40 times along the 400-stimuli sequence. The tone immediately preceding a deviant is the same in the oddball (a standard) and cascaded sequences. The cascaded sequence was recently designed as an improvement to the many-standards, by controlling for the state of refractoriness and the regularity of the deviant tone in the oddball paradigm[35, 36]. This improves the estimation of the overall adaptation state of the system by the time the deviant tone is played, and controls for the potential sensitivity of the neuron to a rise or fall in frequency between two successive tones. Second, the cascaded sequence mimics the regular structure of the oddball sequence, with the important difference that now the target tone conforms to the rule, instead of being a deviant. Thus, using this design, every tone presented as a deviant was also presented as a standard (in a different oddball sequence) and in the context of both the many-standards and cascaded control sequences. These four conditions, and by extension response measures to them, will be denoted as deviant (DEV), standard (STD), many standard control condition and cascade control (CTR). Note that there were two variants of the DEV condition (ascending/descending), which were compared with the corresponding ascending/descending cascade condition. The STD condition was averaged, for each frequency, across ascending/descending versions of the oddball sequence (as indicated in Fig. 1b). The order of presentation of these sequences was randomized across neurons, with a silent pause of ~30 s between sequences. If the neuron could be held for long enough, the same protocol was repeated at different sound intensities.

**Surgical procedures in anaesthetized rats.** Surgical anesthesia was induced and maintained with urethane (1.5 g/kg, i.p.), with supplementary doses (0.5 g/kg, i.p.) given as needed. Dexamethasone (0.25 mg/kg) and atropine sulfate (0.1 mg/kg) were administered at the beginning of the surgery and every 10 h thereafter to reduce brain edema and the viscosity of bronchial secretions, respectively. The initial surgical procedures were identical in each case, and the electrophysiological procedures differed only in the location of the craniotomy, and placement/orientation of the recording electrode, for each different station. After the animal reached a surgical stage of anesthesia, the trachea was cannulated for artificial ventilation and a cistern drain was introduced to prevent brain hernia. The animal was then placed in a stereotaxic frame in which the ear bars were replaced by hollow specula that accommodated a sound delivery system. Corneal and hind-paw withdrawal reflexes were monitored to ensure that a deep anesthetic level was maintained as uniformly as possible throughout the recording procedure. Isotonic glucosaline solution was administered periodically (5–10 ml every 6–8 h, s.c.)

throughout the experiment to prevent dehydration. Body temperature was monitored with a rectal probe and maintained between 37–38°C with a homoeothermic blanket system (Cibertec).

For IC and MGB recordings, a craniotomy was performed in the left parietal bone to expose the cerebral cortex overlying the left IC/MGB. The dura was removed, and the electrode was advanced with an angle of 20° for the IC, and in a vertical direction for the MGB. For AC recordings, the skin and temporal muscles over the left side of the skull were reflected and a 6 × 5 mm craniotomy was made in the left temporal bone to expose the entire auditory cortex (see Fig. 1 in ref. [27]). The dura was removed and the exposed cortex and surrounding area were covered with a transparent layer of agar to prevent desiccation and to stabilize the recordings. The electrode was positioned orthogonal to the pia surface, forming a 30° angle with the horizontal plane, to penetrate through all the cortical layers of one same cortical column.

**Surgical procedures in awake mice.** Experiments in awake mice were performed in 10 CBA/J mice aged between 8 and 12 weeks. Animal handling and surgical procedures for this preparation followed the procedures detailed in previous experiments[65, 66]. Briefly, animals were handled and trained to stay in a customized foam bed, adapted to the animal body, and placed into the stereotactic frame for 5–7 consecutive days. For the initial surgery, anesthesia was induced using a mixture of ketamine (50 mg/kg) and xylazine (10 mg/kg, i.m.). Animals were fixed to the stereotactic frame, skull was exposed, and coordinates for IC or AC (between 2 and 4 mm posterior to bregma, and about 2 mm ventral to linea temporalis), according to refs. [67, 68] were taken. A head-post was implanted as in ref. [66], and a craniotomy was performed, sparing the dura. Analgesic buprenorphine (Buprex™, RB Pharmaceuticals Limited) was injected every 12 h after surgery. The exposed area was protected with a removable silicone elastomer (Kwik-Cast™ & Kwik-Sil™, WPI). At least 3 days after recovery, animals were acclimated to the recording environment with their head and body restrained[65, 66, 68]. Only well-acclimated animals were used to collect data, and mild sedative acepromacine (2 mg/kg, i.p, Equipromacina, Fatro Iberica) was injected in case the mouse showed signs of apprehension during the recordings. Recording sessions were no longer than 3 h, during 2–3 consecutive days.

**Electrophysiological recording procedures.** Each individual animal was used to record from only one auditory station, either IC, MGB or AC. Once a single neuron was isolated and confirmed to be stable, the whole stimulation protocol was applied, as described in the first section "Experimental Design".

Experiments in anaesthetized rats were performed inside a sound-insulated and electrically-shielded chamber. All sounds were generated using an RX6 Multifunction Processor (Tucker-Davis Technologies) and delivered monaurally (to the right ear) in a closed system through a Beyer DT-770 earphone (0.1–45 kHz) fitted with a custom-made cone and coupled to a small tube (12 gauge hypodermic) sealed in the ear.

The sound system response was flattened with a finite impulse response filter, and the output of the system was calibrated in situ using a ¼-inch condenser microphone (model 4136, Brüel & Kjær), a conditioning amplifier (Nexus, Brüel & Kjær) and a dynamic signal analyzer (Photon + , Brüel & Kjær). The output of the system had a flat spectrum at 76 dB SPL (±3 dB) between 500 Hz and 45 kHz, and the second and third harmonic components in the signal were ≤ 40 dB below the level of the fundamental at the highest output level (90 dB SPL). Prior to surgery and recording sessions, we recorded auditory brainstem responses with subcutaneous electrodes to ensure the animal had normal hearing. Auditory brainstem responses were collected using a Tucker–Davis Technologies software (BioSig) and hardware (RX6 Multifunction Processor) following standard procedures (0.1 ms clicks presented at a 21/s rate, delivered in 10 dB ascending steps from 10 to 90 dB SPL).

The experimental procedure for the awake mice was similar to that used for the rats; the main difference was that auditory stimulation in the awake condition was free field (at ~1 cm), presented monaurally to the contralateral ear (the left ear) using an electrostatic loudspeaker (TDT-EC1: Tucker-Davis Technologies) driven by a RZ6 processor. The free field recording was necessary because the mices' heads were immobilized by fixing the head post to a custom-made clamp during recordings. The output of the system at the left ear was calibrated as described above and its maximum output was flat from 1 to 44 kHz (~89 ± 4.3 dB SPL). The highest frequency produced by this system was limited to 44 kHz and the second and third harmonic components in the signal were at least 40 dB lower than the level of the fundamental at the highest output level[65].

Action potentials and local field potentials were recorded with hand-manufactured, glass-coated tungsten electrodes (1–4 MΩ impedance at 1 kHz). One individual electrode was used to record one single neuron at a time. The electrode was advanced using a piezoelectric micromanipulator (Sensapex) until we observed a strong spiking activity synchronized with the train of searching stimuli. The signal was amplified (1000×) and band-pass filtered (1 Hz to 3 kHz) with an alternate current differential amplifier (DAM-80, WPI). This analog signal was digitized at a 12 K sampling rate and further band-pass filtered (with a second TDT-RX6 module) separately for action potentials (between 500 Hz and 3 kHz) and LFP (between 3 and 50 Hz). Stimulus generation and neuronal response processing and visualization were controlled online with custom software created

with the OpenEx suite (Tucker-Davis Technologies) and Matlab (Mathworks). A unilateral threshold for automatic action potential detection was manually set at about 2–3 standard deviations of the background noise. Spike waveforms were displayed on the screen, and overlapped on each other in a pile-plot to facilitate isolation of single units. Only when all spike waveforms were identical and clearly separable form other smaller units and the background noise, the recorded action potentials were considered to belong to a single unit.

To confirm that our recordings corresponded to well-isolated single units, we used 2552731 individual spike waveforms from 5871 record files from all stations to measure spike isolation quality. Inter-spike interval distribution for all recorded spike waveforms (Supplementary Fig. 3a) shows that only 0.85% spikes occurred less than 4 ms after the previous spike. To show that waveform variability was low in our recordings (as indicated by the sample spike waveform in Fig. 2), and that spike amplitude was well above background noise level, we computed a spike-amplitude-to-noise-ratio (SNR), for all sets of spike waveforms S recorded:

$$\text{SNR} = \frac{\max(\bar{x}(\mathbf{S})) - \min(\bar{x}(\mathbf{S}))}{\text{Std}(\mathbf{S})}$$

Spike-amplitude-to-noise-ratio distribution in our sample (Supplementary Fig. 3b) shows that 96% of our recorded spikes had at least 5 times more amplitude than the background noise, and that 61% of them were well above 10 times that. Finally, to ensure that all spike waveforms of every record belonged to a single neuron, Mahalanobis distance was computed for each of them. Mahalanobis distance is a normalized measure separation between a point and a cluster of point in a multidimensional space. If more than two neurons were recorder together, the spike waveforms would follow a multimodal Gaussian distribution, and the median Mahalanobis distance would be larger than for a single Gaussian distribution. Our spike waveforms were streams of 32 samples (5 ms at 12 K sampling rate); thus, in our case, Mahalanobis distance is a normalized measure of separation between a spike waveform and a cluster of spike waveforms in a 32-D space. If our spikes were purely normally distributed following a single 32-dimensional Gaussian distribution, the distribution of mahal (w, S) values for all spike waveforms w would look like the red dotted line in Supplementary Fig. 3c. However, the real distribution (blue histogram) is a left-skewed version of the former, indicating that our spike waveforms were even closer to each other in shape than in a standardized single-spike cluster (Supplementary Fig. 3d, e).

**Histological procedures and localization of recording sites.** For AC experiments, a magnified picture (25×) of the exposed cortex was taken at the end of the surgery with a digital single-lens reflex camera (D5100, Nikon) coupled to the surgical microscope (Zeiss) through a lens adapter (TTI Medical)[27]. The picture included a pair of reference points—previously marked prior surgery on the dorsal ridge of the temporal bone - indicating the absolute scale and position of the image with respect to bregma. This picture was displayed on a computer screen and a micrometric grid was overlapped to guide and mark the placement of the electrode for every recording made (Supplementary Figs. 4a and 5a). Recording sites (250–500 μm spacing) were evenly distributed across the cortical region of interest and avoided blood vessels. The vascular pattern was used as a local reference to mark the position of every recording site in the picture, but otherwise differed between animals.

At the end of the experiment, the limits and relative position of the auditory fields were determined for each animal. This was done using the characteristic frequency, the tone frequency that elicits a significant neuronal response at the lowest intensity gradient, as the main reference landmark. Five auditory cortical fields were identified according to tone frequency-response topographies both in rats[27, 69] and mice[67, 68]. In rats, we consistently observed distinct tonotopic gradients within the different fields with a high-frequency reversal between ventral and anterior auditory field (rostrally), a low-frequency reversal between primary and posterior auditory field (dorsocaudally) and a high-frequency reversal between ventral and suprarhinal auditory field (ventrally) (Supplementary Fig. 4a). We identified the boundary between primary and ventral auditory field as a 90° shift in the characteristic frequency gradient in the ventral low-frequency border of primary auditory cortex, and the boundary between primary and anterior auditory field as an absence of tone-evoked responses in the ventral, high-frequency border of primary auditory cortex[27]. We used these boundaries to assign each recording to a given field. The characteristic frequency of each recording track was computed as the average characteristic frequency of all neurons recorded in that track, including a fast multi-unit activity frequency-response area recording made between 400–550 μm depth, corresponding to layers IIIb-IV of the AC.

Similar tonotopic gradients were observed in mice (Supplementary Fig. 5a) in accordance with previous studies[67, 68]. Inversions of the characteristic frequency progression define the limits between cortical fields[67, 68] so that most recordings could be assigned to a particular field: primary, secondary, dorsal posterior, or insular auditory field. Tonotopic maps were less distinct in mice because the mice data sample was smaller than in rats. Furthermore, since mice AC is smaller, mappings are less detailed than those in rat.

For IC and MGB experiments, each recording track was marked with electrolytic lesions for subsequent histological localization of the recorded neurons. At the end of the experiment, the animal was given a lethal dose of sodium pentobarbital and perfused transcardially with phosphate buffered saline (0.5% NaNO₃ in Phosphate Buffered Saline) followed by fixative (a mixture of 1% paraformaldehyde and 1% glutaraldehyde in rat Ringer's solution). After fixation and dissection, the brain tissue was cryoprotected in 30% sucrose and sectioned on a freezing microtome in the transverse or sagittal planes into 40 μm-thick sections. Sections were Nissl stained with 0.1% cresyl violet to facilitate identification of cytoarchitectural boundaries (Supplementary Figs. 4b–e and b, c). Recording sites were marked on standard sections from a rat/mouse brain atlas[70, 71] and neurons were assigned to one of the main divisions of the IC (central nucleus, dorsal, lateral or rostral cortex) or the MGB (ventral, dorsal and medial division), respectively. The stained sections with the lesions were used to localize each track mediolaterally, dorsoventrally and rostrocaudally in the Paxinos atlas. To determine the main IC or MGB subdivisions, cytoarchitectonic criteria, i.e., cell shape and size, Nissl staining patterns and cell packing density were used. This information was complemented and confirmed by the stereotaxic coordinates used during the experiment to localize the IC/MGB. After assigning a section to each track/lesion, the electrophysiological coordinates from each experiment and recording unit, i.e., beginning and end of the IC/MGB, as well as the depth of the neuron, were used as complementary references to localize each neuron within a track.

**Statistical analysis.** All the data analyses were performed with the Matlab™ software, using the built-in functions, the Statistics and Machine Learning toolbox, or custom scripts and functions developed in our laboratory. A peri-stimulus histogram was a histogram of action potential density over time (in action potentials per second, or Hz) from −75 to 250 ms around stimulus onset, using the 40 trials available for each tone and condition. Every peri-stimulus histogram was smoothed with a 6 ms gaussian kernel ("ksdensity" function in Matlab) in 1 ms steps to estimate the spike-density function over time, and the baseline spontaneous firing rate was determined as the average firing rate (in Hz) during the 75 ms preceding stimulus onset. Peri-stimulus histograms were generated for each stimulus/condition tested. Only the last STD tones preceding each DEV tone were used for the analyses. The excitatory response was measured as the area below the spike-density function and above the baseline spontaneous firing rate, between 0 and 180 ms after stimulus onset (positive area patches only, to avoid negative response values). This measure will be referred to as "baseline-corrected spike count".

We only analyzed excitatory responses, since we look primarily for enhancement of responses to deviant tones. Neuron/frequency combinations with no significant excitatory response to at least one of the conditions (DEV, STD, CTR) were excluded from the analyses (p > 0.05 for all three conditions). To test for statistical significance of the baseline-corrected spike count, we used a Monte Carlo approach, a probability simulation that obtain numerical results from several random sampling. First, 1000 simulated peri-stimulus histograms were generated using a Poisson model with a constant firing rate equal to the spontaneous firing rate. Then, a null distribution of baseline-corrected spike count was generated from this collection of peri-stimulus histograms, following these same steps. Finally, the p value of the original baseline-corrected spike count was empirically computed as $p = (g + 1)/(N + 1)$, where g is the count of null measures greater than or equal to baseline-corrected spike count and $N = 1000$ is the size of the null sample.

We used two types of sequences to control for repetition effects namely the many-standards and cascaded sequences (Fig. 1b). However, it is possible to decompose the neuronal mismatch into repetition suppression and prediction error using either of these sequences alone (Fig. 1c). Here we describe the analysis performed using the cascade condition as control (CTR), since the analysis using the many-standards sequence was completely analogous. Baseline-corrected spike count responses of a neuron to the same tone in the three conditions (DEV, STD, CTR) were normalized using the formulas:

$$\text{DEV}_{\text{Normalized}} = \text{DEV}/N;$$

$$\text{STD}_{\text{Normalized}} = \text{STD}/N;$$

$$\text{CTR}_{\text{Normalized}} = \text{CTR}/N;$$

where

$$N = \sqrt{\text{DEV}^2 + \text{STD}^2 + \text{CTR}^2}$$

is the Euclidean norm of the vector (DEV, STD, CTR) defined by the three responses. This normalization procedure always results in a value ranging 0–1, and has a straightforward geometrical interpretation (Fig. 3a): Normalized values were the coordinates of a 3D unit vector (DEV_Normalized, STD_Normalized, CTR_Normalized) with the same direction of the original vector (DEV, STD, CTR), and thus the same proportions between the three response measures. From these normalized responses, indices of neuronal mismatch (iMM), repetition suppression (iRS), and

prediction error (iPE) were computed as:

$$iMM = DEV_{Normalized} - STD_{Normalized},$$

$$iRS = CTR_{Normalized} - STD_{Normalized},$$

$$iPE = DEV_{Normalized} - CTR_{Normalized},$$

These indices, consequently, always range between −1 and 1, and provide the following quantitative decomposition of neuronal mismatch (Fig. 1d) into repetition suppression and prediction error:

$$iMM = iRS + iPE$$

As shown in Supplementary Fig. 2, the iMM was largely equivalent to the typical "SSA index", commonly used in most previous studies of SSA in single units[29, 37]:

$$SSA\ index = (DEV - STD)/(DEV + STD)$$

For the analysis of the LFP signal, we aligned the recorded wave to the onset of the stimulus for every trial, and computed the mean LFP for every recording site and stimulus condition (DEV, STD, CTR), as well as the "prediction error potential" (PE-LFP = $LFP_{DEV} - LFP_{CTR}$). Then, grand-averages were computed for all conditions, for each auditory station separately. The $p$ value of the grand-averaged PE-LFP was determined for every time point with a two-tailed $t$ test (Bonferroni-corrected for 200 comparisons, with family-wise error rate FWER < 0.05), and we computed the time intervals, where PE-LFP was significantly different from zero (Fig. 5).

Our data set was not normally distributed so we used distribution-free (non-parametric) tests. These included the Wilcoxon signed-rank test and Friedman test (for baseline-corrected spike counts, normalized responses, indices of neuronal mismatch, repetition suppression and prediction error). Only the difference wave for the LFPs (PE-LFP in Fig. 5) was tested using a $t$ test, since each LFP trace is itself an average of 40 waves, and thus approximately normal (according to the Central Limit Theorem). For multiple comparison tests, $p$ values were corrected for false discovery rate (FDR = 0.1) using the Benjamini-Hockberg method. Linear models used to test significant average iPE within each auditory station (Fig. 4b, d) and significant effects of nucleus, hierarchy, SPL, direction, and interactions between them, were fitted using the 'fitlm' function in Matlab, with robust options. To estimate final sample sizes required for the observed effects after the initial exploratory experiments, we used the 'sampsizepwr' function in Matlab. The central measure of this study was the iPE, and thus we adjusted sample sizes, for each station, to obtain a statistical power of 0.8 for this index, given the observed effect:

MinSampleSize = sampsizepwr('t',[0 std(iPE)],max(.05,abs(mean(iPE)),.8);

where iPE is the distribution of iPE values in the sample, including all frequencies tested for all neurons ("points" in Table 1). Sample sizes were enlarged with additional experiments until they were just greater than the minimum required (number of points recorded and the minimum required for each station: $IC_L$ = 149/ 104; $IC_{NL}$ = 523/401; $MGB_L$ = 79/69; $MGB_{NL}$ = 211/153; $AC_L$ = 250/125 and $AC_{NL}$ = 307/29). In some cases, such as $AC_{NL}$, final sample sizes were much larger than required (307 points recorded for 29 required), due to four very productive experiments.

**Code availability**. The scripts and functions written in Matlab to generate the results and analysis during the current study are available from the corresponding author on reasonable request.

**Data availability**. The data sets generated and analyzed during the current study are available from the corresponding author on reasonable request.

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

## Acknowledgements

We thank Drs. Ryszard Auksztulewicz, Edward L. Bartlett, Nell Cant, Javier Cudeiro, Bernhard Englitz, Yonatan Fishman, Patrick May, Alan Palmer, José Luis Peña, Daniel Polley, Adrian Rees, Iria SanMiguel, Christoph Schreiner, and Juanita Todd for their comments and many fruitful discussions on previous versions of the manuscript and for their constructive criticisms. Financial support was kindly provided by the Spanish MINECO (SAF2016-75803-P) to MSM and an Explora-Ciencia grant (PSI2013-49348-EXPLORA) to MSM and CE. CE was also supported by the Generalitat de Catalunya (SGR2014-177) and by the Icrea Acadèmia Distinguished Professor Award. JND held a fellowship from the European Social Fund/Spanish JCYL (Operational Programme ESF Castilla y León 2007–2013). GGP held a fellowship from the Spanish MINECO (BES-2014-069113) and CVB held a fellowship from Mexican CONACyT (216652).

## Author contributions

The experiments were performed at the Auditory Neuroscience Laboratory, Institute of Neuroscience of Castilla y León- INCYL, University of Salamanca, Salamanca, Spain. The contribution of each author to the following aspects of the study is as stated: (1) conception and design of experiments: J.N.D. and M.S.M.; (2) collection of data: J.N.D., G.G.P., G.V.C. and C.V.B.; (3) analysis, interpretation of data and conceptual advice: J.N.D., G.V.C.; G.G.P.; C.E. and M.S.M.; (5) writing of the manuscript: J.N.D. and M.S.M. (6) project supervision: M.S.M. All authors approved the final version of the manuscript.

## Additional information

**Competing interests:** The authors declare no competing financial interests.

