## [Peer Review File · Nature Communications]

Reviewers' comments:

Reviewer #1 (Remarks to the Author):

Nieto-Diego and colleagues investigate the neural correlates of mismatch negativity. I would recommend publication. The results are topical and interesting – my main reservation is the presentation and the pitch of the results. This is detailed in the following:

1. Please rephrase abstract, introduction and discussion. Overall the text lacks a clearly structure introduction with a comprehensive review of previous results (e.g. the results of Taaseh et al. 2011, or Hershenhoren 2014, that both use very similar methods in A1 and come to different conclusions are not discussed). Conversely, in relation to interpretation of SSA as a predictive mechanism, the problem is that the authors imply throughout the manuscript that predictive coding is only used to describe MMN effects (e.g. line 29ff: “we demonstrate that the predictive activity of sensory systems is detectable at the neuronal level” – or line 228: “evidence, hitherto unavailable”)? Single neuron prediction error recordings are quite abundant, and “predictive activity” has been demonstrated in dopaminergic neurons (e.g. Schultz et al. 1997), the cerebellum (e.g. Ohmae & Medina 2015), auditory feedback monitoring in auditory cortex (e.g. Eliades & Wang 2008, Keller & Hahnloser 2009, Schneider et al. 2014), deviance detection in S1 (Musall et al. 2015), and even mismatch processing in the visual system (Meyer & Olson, Zmarz & Keller 2016 – as the authors highlight in the discussion, but also Rao & Ballard 1999, Keller et al. 2012, Saleem et al. 2013, or Roth et al. 2016 for thalamic prediction error signals).

The authors could structure their introduction to highlight that the concepts of predictive coding are quite successful and frequently used to describe brain function, and that these same concepts can also be applied to describe SSA or MMN (as has also been argued previously).

Also, in many places the clarity of the manuscript could probably be improved, e.g.:

- Line 47f “seamlessly fitting it as the sum of thousands of neuronal predictions error signals”: not only are there grammatical problems in this sentence (e.g. what does ‘it’ refer to), but what the authors are trying to say is unclear (possibly the latter is a consequence of the former)?

- “higher stations” is likely better described as “higher levels” (of the processing hierarchy).

- Line 54ff: “neuronal response progressively switch from representing the stimulus itself to represent the sensory prediction error”. Although I think I understand what the authors mean here, this simplification is wrong. In predictive coding, top down signals encode predictions, feed-forward signals encode prediction errors, there is no “switching” or even “representation of the stimulus” (technically, everywhere but in the first level of the hierarchy, top-down signals predict the feed-forward prediction error, not a stimulus representation in the classical sense)?

- Line 55ff: "This is why..." is a bit strong here given that the authors describe a theory.

- Line 87f: "the role of subcortical structures in cognition [...] remains largely unexplored". Absolutely correct, but irrelevant given that also this manuscript does not address the role of subcortical structures in cognition.

- Line 142f: seems unnecessarily aggressive.

2. If the authors want to highlight the relevance of their own data to MMN experiments in humans, they should add a thorough discussion of the influence of the anesthesia used in their experiments.

3. There are a few apparent inconsistencies in the figures/tables:

- Figure 3b: in the panel the comparison is labeled as n.s., but in table 1 it is reported as $p=0.019$ with a negative change. This just means the DEV response is significantly smaller than the CAS response, not that it is not significant?

- Figure 3e: all neurons/tones should appear in all three "walls" – this is not the case here. In addition, it is very hard to believe – based on the data shown in lower right wall (DEV-CAS) – that the distribution of points is on the right of the diagonal (i.e. stronger for the DEV, as table 1 would indicate)?

- Figure 5c: there is almost no visible difference in the LFP DEV and CAS responses during the first negative peak, but the PEP is clearly negative?

- In addition, in most figures there are axis and line labels missing (e.g. Figure 2 y-axis has only one number, Figure 4a x-axis label is missing, etc. etc.)

Minor suggestions for improvement:

4. Line 43ff: This is misleading at best – a biomarker with a d' of roughly 1 (see e.g. Light and Swerdlow 2015) can hardly be referred to as a "reliable" biomarker (d' of 1 means e.g. at a 84% hit rate a 50% false positive rate). Please remove "reliable".

5. Line 175: please either add the IC terms or explain why you are not listing them. Also, use of only "ho" in the equation (without using "fo") could be considered confusing.

6. Line 179ff: Finding a significant effect of one of the variables in a model fit does not imply that all comparisons along that variable are significant. Please rephrase.

7. Line 188ff: Why is there an asymmetry between ascending and descending deviants?

8. Line 362: "surgical plane of anesthesia" should probably be "surgical stage of anesthesia" (typically, the surgical stage is divided into four planes)?

9. Line 683: Figure 1e. I would suggest to label the three different types of differences as DEV-CTR (instead of prediction error), STD-CTR (instead of repetition suppression), etc. The authors could then state that under the assumption of the predictive coding hypothesis, DEV-CTR would correspond to prediction error, etc. The problem is that the schematic

currently implies the existence of negative prediction error.

10. Figure 3: I suggest either showing the raw data or the normalized data, but not both

Reviewer #2 (Remarks to the Author):

Summary

The manuscript by Nieto-Diego et al. present an interesting body of findings on how novel or deviating stimuli are encoded and detected along the auditory pathway on the level of individual neurons in a predictive coding framework. To approach this question they recorded extracellular activity in different auditory centres (inferior colliculus IC, thalamus MGB and auditory cortex AC) and presented oddball stimulus sequences to the anesthetized rat. In order to disentangle rather simple adaptive processes (e.g. stimulus-specific adaptation, repetition suppression) from predictive activity (or true deviance detection) they use different carefully designed control conditions. These conditions allow controlling for different effects such as the rarity of a stimulus and regularity of a sequence. Neurons exhibiting predictive activity should respond to a deviating stimulus embedded in an oddball sequence with increased firing rate but with a lower activity to the same stimulus when presented in one of the control conditions.

The central finding of their work is that there is increase in predictive activity along the ascending auditory pathway, even in subcortical structures. This is the first time that predictive single neuron activity has been systematically tracked and demonstrated along an ascending sensory pathway by employing a set of rigorous controls that are essential for a convincing proof. Interestingly, predictive activity evolves gradually along the different stages implying that each auditory centre contributes to its computation or at least amplification. Interestingly, even higher order subcortical structure exhibit predictive activity.

This manuscript is a very comprehensive and important body of work which uses the appropriate methodology and analysis. The work will be of interest to a wider audience, in particular one that is focusing on predictive coding and mismatch negativity (MMN) in human subjects. The understanding of the neuronal basis of MMN is still very superficial despite its extensive usage in studying central sensory processing and pathologies in humans. Due to its comprehensive nature, this paper will certainly be extremely helpful for a broad readership

Major comments

Having said this, I have a couple of concerns regarding the figures, methods, results and conclusions.

1. Figures: In general, the figures are not very carefully made. While I understand the urge to make figures concise, it leaves the reader guessing what the axes and scales exactly are. I highly recommend reworking carefully every figure. Just some (incomplete) examples:

- Figure 2a: only some x- and y-axis are labelled properly.

- Figure 2b: there are no axis- or tick labels at all. This is just not acceptable. Even the bar indicating the stimulus in the last row is missing. There is more than enough space to label them properly
- Figure 3 a-f: There are no tick-labels at all for the scatterplot, is it ranging from -1 to 1?
- Figure 4 a, b and c. Only in panel b the first order (fo) and higher order (ho) is indicated for the inferior colliculus. Is it the same order to the other areas? This has to be labelled properly. In panel d the y-axis is not labelled.
- Figure 5: No time axis label at all in any panel. Most y-labels are missing as well.

2. Methods: The authors claim to provide data from single units. However, the only proof of this claim is the spike waveforms in figure 1A and figure 2. This is not enough. The spike detection threshold was set to 2-3 standard-deviations of the noise floor (page 17, line 25). This is a comparatively low threshold and combined with rather low electrode impedance (1-4 MegaOhm, page 17, line 15) this will typically yield multi-unit activity and not single units. Therefore, measures of the spike quality (distribution of inter-spike-intervals, real signal-noise ratio, amplitude in μV , Mahalanobis distance, etc. etc.) have to be provided and ideally spike-sorting (even without tetrode) should be performed in order to convince the reader. This leads to another comment that may improve the quality of the manuscript. When looking at the spike waveforms in figure 2A, third row (insets) it seems that there are significant differences in spike-width between units. So probably, it makes sense to analyse the data with respect to the spike width (fast- vs. regular spiking). While it is debatable, if the spike-width really corresponds to inhibitory or excitatory neurons it would be very interesting to check and see if the findings of Chen, Helmchen & Lütcke 2015 (Journal of Neuroscience) can be confirmed for some auditory areas.

3. Methods: The anaesthesia was set to "moderately deep" (page 16, line 7) for the recordings but "deeply anaesthetized" prior to surgery with urethane (page 12, line 19). I wonder how the authors were able to switch from deep to moderately deep anaesthesia with urethane which is very difficult to control. In general, anaesthesia is a concern. Previous studies in awake animals (e.g. von der Behrens et al. 2009, Journal of Neuroscience; Farley et al. 2010, Journal of Neuroscience) did not find a dramatic change in the effect size of stimulus-specific adaptation compared to similar studies in anaesthetized animals (e.g. Taaseh et al. 2011, Plos One) where the SSA effect seems to be slightly stronger. Furthermore, the typical predictive signal recorded in humans as MMN can be measured under anaesthesia as well. However, Harms et al. 2011 (Front Psychol) saw differences in MMN signals in the rat under anaesthesia compared to the awake condition. And given the fact that the authors are carefully disentangling effects of simple adaptation and real predictive signals it could be very interesting to have a look at the awake condition as well. This would certainly strengthen the manuscript significantly.

4. Methods: Histology. Apparently, the authors did a histological confirmation of their recordings sites (page 19 ff). However, the proof of this is lacking. This is even more important, as for some regions it is very difficult to draw boundaries between different regions in the subcortical structures. I recommend providing at least examples of histological preparations together with boundaries as supplementary material.

5. Methods: The stimulus repetition rates were different for different brain regions (AC: 3

Hz, IC & MGB: 4Hz, page 14, line 2). This is problematic, as it is well known that effects of stimulus-specific adaptation, repetition suppression and predictive coding depend massively on the repetition rate. Therefore, the effect size in cortex is most likely underestimated compared to the subcortical structures (or vice versa). I would recommend choosing just the higher repetition rate (4 Hz) so that each structure is properly driven by the stimulus. The whole evolution of the effect should become clearly by employing this strategy.

6. Methods: As one can see in figure 2, first row, the tuning curves of different units vary significantly. It is known that tuning curves differ between different auditory areas. In general, the tuning in the central nucleus of the IC is sharper than for example in the thalamus or cortex. This leads to two questions. First, the authors should provide measures of the tuning-curves in the different structures (in particular for the width Q10dB or Q20dB). This is important because for broadly tuned neurons (as in AC or MGB) there are more stimuli of the control sequences (CAS and MAS) within the tuning curve than for narrowly tuned neurons in the IC where probably only one or two stimuli of the cascade fall within the tuning curve (see figure 2A, first row). The key measure for predictive activity is the response difference between tones embedded in an oddball sequence compared to the same tone embedded in a cascade sequence (CAS). When there is less cross-frequency adaptation in the cascade condition for the IC units one may expect only a small difference between the oddball deviant and the cascade. Currently, the authors do not find predictive activity in the first order IC. However, when they customize the control CAS sequences to the tuning curve width, as they already do for the frequency and amplitude, they may find a different result. At the very least they should look for the correlation between tuning curve width and effect size.

7. General: A central finding of the manuscript is the subcortical existence of real predictive activity: “[...] and that this predictive activity emerges hierarchically from subcortical structures” (page 4, line9-10, see also page 2, line 9, page 8, line 3-4, page 8, line 17-20 and page 10, line 16). However, according to figure 4B (yellow bars) and figure 5D & E, predictive activity can be only observed in higher order subcortical structures (ho-MGB and very weakly in ho-IC). These structures receive massive corticofugal projections. Therefore, it is quite misleading to suggest that predictive activity emerges hierarchically from subcortical structures. It is a very likely scenario that the predictive activity is actually projected back from the auditory cortex to MGB and IC. This should be clearly stated in the discussion and in general toned down.

8. Discussion: The authors find an interesting asymmetry between the ascending and descending cascade stimulus sequence (CAS, figure 4C). In the discussion (page 12, line 4-5) the authors state that “asymmetries in the direction of frequency-change detection (ascending vs. descending) have also been found in both animal (Harms et al. 2014) and human MMN studies (Peter et al. 2010)”. In contrast to the two cited ERP studies, Klein et al 2014 (Brain Topog.) find in their awake single- and multi-unit recordings only a weak asymmetry of cortical stimulus-specific adaptation (SSA) to frequency modulated tones (FM) which can be comparable to the cascade stimulus sequence used in the current manuscript. This should be taken into account.

9. Figure 3 A-I: The 3D projections (insets) are very small and hard to read. I don't think they are particularly helpful for understanding the message of this figure. They just complicate the figure unnecessarily. Panel G-I: The key information is in the histograms for iRS, iMM and iPE. Therefore I would just show the histograms for the three conditions and improve their readability. For example one could plot the histograms for one condition (e.g. iPE) below each other following the brain hierarchy. This will facilitate the comparability between distributions of different areas (for one condition) and therefore illustrate the main claim of the manuscript even better ("Hierarchical Prediction Error")

10. Discussion: The authors claim that "Previous attempts to show predictive activity in auditory neurons were inconclusive" (page 11, line 5-6). However, in the next sentence the authors cite two papers that demonstrate exactly this for the auditory and barrel cortex. Furthermore, the authors do not cite a paper by Hamm & Yuste 2015 (Cell reports) who find MMN-like activity in the visual cortex of the awake mouse. This is important because Hamm & Yuste employed the many standards control (MAS) as well demonstrating MMN-like activity in the LFPs and single neurons (2p Calcium imaging) and that it was dependent on the somatostatin-containing interneurons. Therefore, the literature addressing predictive activity in single neurons is not inconclusive anymore. In particular as the authors themselves do not see a big difference between the many-standards control MAS (used in the three references mentioned above) and the cascade control CAS.

Minor comments

1. Page 13, line 5: "relays" instead of "relies".
2. References to figure panels are in upper-case but the panels themselves are written in lower-case. This should be done uniformly.
3. Figure 4B: what is the interpretation of the negative prediction error (hashed area) in the first-order MGB?
4. Methods, page 20, line 22-23: "[...] to avoid negative response values". Why were negative values excluded? In particular cortical neurons exhibit a lot of inhibition during sustained stimulation, even during the 75 ms stimulus duration used here. I don't see the reason to exclude these periods. And was the excitatory response calculated for the time window 0 to 250 ms? On the other hand according to figure legend 2, second row (2) the activity was averaged for all units during 180 ms. A blanket averaging approach for all conditions and brain areas could skew the result significantly as well. This whole procedure should be explained in more detail and the authors have to take into account that neurons in different brain areas exhibit different activity patterns.
5. Discussion, page 12, line 6: "[...] demonstrate that prediction error is an intrinsic component of responses of single auditory neurons". What do the authors mean by "intrinsic" in this context? A prediction error activity is very unlikely the result of a solely intracellular process but more likely the result of network activity. This should be clarified.
6. Figure 3G and figure 4B: in figure 3G the median iPE is slightly negative (yellow line at yellow distribution for fo-IC). However, in Figure 4B, the iPE for fo-IC is slightly positive. Where does this mismatch come from?
7. Figure 4 legend: Typo "represented".

8. Table 1: what does a p-value of 0.000 correspond to?

Reviewer #3 (Remarks to the Author):

The study by Nieto-Diego et al. is a highly complex study on various stages of the auditory system in deeply anesthetized rats using extracellular single-neuron electrophysiology (as well as local field potentials) and various types of tone sequences for auditory stimulation. The study pursues the ambitious goal to “unify three coexisting views of perceptual deviance detection at different levels of description: neuronal physiology, cognitive neuroscience and the theoretical predictive coding framework”. Put more simply, the study strives to establish an animal model of mismatch negativity (MMN) and to test alternative mechanisms as its basis: predictive coding (or Bayesian inference) versus stimulus-specific adaptation (SSA). The authors conclude that “prediction error is an intrinsic component of responses of single auditory neurons”, including in subcortical centers, and forms the basis of human MMN. They hope that this will lead to an understanding of the cellular basis of schizophrenia.

I found the manuscript extremely hard to read and the arguments quite difficult to follow. The complexity of the study lies not only in its design: Cells were recorded from six different brain structures in 36 animals, and were tested with six different tone sequences consisting of 400 (!) stimuli, after determining the frequency response area of a neuron at different intensities. This must take an enormous amount of time and raises concerns about recording stability.

The authors also make a number of implicit theoretical assumptions about the organization of the auditory system (and sensory systems in general), which are not straightforward and are not well explained. Some of these assumptions, e.g. that there are “higher-order divisions” of the MGB, are tucked away in the Methods section (p. 13), where they are hard to find. These assumptions do not conform with the current understanding of auditory cortex organization in other species.

Although the total number of cells tested is above 200, which seems like an adequate number overall, this number dwindles to 36 or less in 5 of the 6 structures tested. These are actually quite small sample sizes (Table 1), rendering statistical conclusions less powerful. One of the main conclusions is based on the statistical difference between “first-order” and “higher-order” neurons, which might require larger sample sizes for the effects to hold up. Furthermore, for the results from “higher-order” structures on each level to be collapsed one has to assume that the neuronal responses recorded in these regions are homogeneous, which is almost certainly not the case (see below). Furthermore, neurons in nonprimary auditory cortex are notoriously hard to drive with pure tones (including presumably pure-tone sequences), so determination of an FRA may not always be reliably possible, which jeopardizes subsequent analysis.

One could also argue that recordings in anesthetized animals make the results relatively less convincing, since the ultimate conclusions are claimed to be about basic mechanisms of cognition. Recordings in awake animals would seem to be preferable.

Another possible shortcoming is that, although the authors performed histological reconstructions of their electrode tracks, which were used to assign the recorded cells to different structures, they did not report recording depths of their cortical neurons. Thus they are unable to provide information on layer-specific effects in auditory cortex, which may be important, as there is classical evidence that neurons in supragranular and infragranular layers of cortex play different roles in feedforward and feedback processing (Van Essen; Rockland & Pandya; Bullier & Kennedy; and others). Admittedly, the thinner extent of cortex in rodents than in primates makes it harder to determine laminar specificity.

The authors reference the studies of Lee and Sherman (2011) as their model for the organization of ascending pathways into “first-order” and “higher-order” structures at every level. According to this model, different subnuclei in the inferior colliculus (IC) and the medial geniculate body (MGB) are at different hierarchical levels, hence the terms “first-order” and “higher-order”. A competing notion is that these subnuclei are organized in parallel and mediate different functions. In the barn owl, for instance, a central nucleus (ICC) is responsible for tonal or frequency information, whereas an external nucleus (ICX) is important for spatial hearing. The same assumption of functional subdivisions and parallel processing streams has been made for the auditory system of higher mammals, including humans and nonhuman primates (Kaas, Hackett, and others): Although the overall organization is obviously hierarchical from brainstem to cortex, there are multiple parallel stages at each level performing different functions, beginning with a dual architecture at the level of the cochlear nucleus (VCN and DCN). This parallel-hierarchical organization is epitomized at the cortical level, where multiple functionally specialized areas are assumed to work in parallel (in addition to forming several hierarchical levels). Multiple primary-like areas exist as well. The assumption of just one “information-bearing” structure at every level (“fo”) and one higher-order (modulatory) structure (“ho”) seems too simplistic.

In summary, it would be more effective to concentrate on the most interesting finding which, in my opinion, is that auditory cortex is in fact substantially different from subcortical structures (see Fig. 3 c,f) (though the authors emphasize a different view), and that nonprimary auditory cortex, in particular, is different from all other structures in that it shows by far the clearest signs of predictive coding (Figs. 4b,d and 5f). Whether this is indeed the neural correlate of MMN (and whether this has any relevance for schizophrenia) remains to be seen until the electrophysiological findings can be compared with behavioral results in the same species. More specificity for different brain regions in, e.g., temporal and frontal cortex would also have to be demonstrated.

Reviewer #4 (Remarks to the Author):

Nieto-Diego et al provide evidence of a hierarchical organisation of prediction errors at a neuronal level in the ascending auditory system: they observed minor levels of predictive processing in IC, but in higher order IC areas only, similarly in MGB but at an increased level relative to IC and substantial predictive error in AC in both first and higher order AC

areas. The importance of this finding is that these observations are at the neuronal level rather than at large scale network level such as exemplified by mismatch negativity (MMN) recorded from scalp recordings and clearly provide evidence of hierarchical organisation of increasing predictive error from subcortical areas to cortical areas. The design of the study is exemplary in that the authors have used optimal control conditions to separate predictive error effects from repetition suppression or stimulus specific adaptation (SSA) effects. While I have queries or require clarification about some conclusions or Figures, the findings will be of considerable interest to neuroscientists in general.

Predictive coding frameworks are increasingly being adopted as models of brain function following Friston's seminal publications. Repetitive patterns of stimulation are proposed to generate internal models of the environment that enable the brain to anticipate its next state of activation. When sensory input produces an activation pattern that matches model predictions, prediction-error is largely suppressed, thus assisting the brain to conserve resources when sound provides no new information. However, change in the environment leads to the elicitation of prediction-errors signalling that an update to the model may be required. The somewhat cortico-centric literature on predictive coding to date implicates a cortical hierarchy whereby the predictions of the model are expressed in top-down (higher to lower areas) connections that control the responsiveness to sensory input. Responsiveness is reduced in cortical regions encoding the predicted activation (e.g., the neurons sensitive to the attributes of the predicted sound), but it is increased in alternate cortical regions (sensitive to non-predicted attributes). The discrepancy generates an error-signal communicated bottom-up (lower to higher areas). What Nieto-Diego et al have shown is that predictive error coding begins at midbrain levels. All areas investigated by Nieto-Diego et al showed evidence of response suppression (or what others have referred to as adaptation or SSA) that is particularly marked at sub-cortical areas but as one moves up the auditory ascending pathway, predictive error processing becomes dominant. The paper will therefore have a strong influence on the field and lead to further research in sub-cortical nuclei and the feed-forward/feedback mechanisms that account for critical features of predictive coding.

As noted earlier, the methodology is exemplary – in terms of stimulus control, responses to same physical stimulus (same intensity and frequency) when it is a standard (SDT) in an oddball sequence or a deviant (DEV) or a control (CTR) stimulus in a cascade (CAS) or many-standards (MAS) sequence are always employed), and the distance between each of these stimulus types is kept fixed at 0.5 octave (in terms of frequency). The use of the cascade control sequences is very clever and confirms my suspicion that the reason the cascade sequence did not work (in that it did not confirm evidence of true deviance detection) in surface recordings in the rat is because the equivalent of the oddball deviants were placed at the extremes of the ascending and descending cascades (see Harms et al., 2014) and therefore probably as deviant as an oddball deviant. Nieto-Diego et al's decision to place the equivalent of the deviant in the middle of either an ascending or descending sequence (although not really equivalent to the original cascade sequence as used by Ruhnau et al 2012) is sensible – it controls not only for background regularity and probability but also for the frequency (Hz) of the preceding stimulus (which is equivalent to the standard of an oddball sequence). Of course if rodents are unable to model the

regularity (either an ascending or descending sequence of sounds) and we have no data that I know of on this issue, then the cascade sequence presumably acts in much the same way as the many standards control. The authors state that the outcomes of their analyses are much the same whether they use cascade or many standards control sequences for comparison purposes with DEV (or STD) responses but inspection of Table 1 suggests that the cascade control may have provided a more sensitive measure of prediction error (iPE) than the many standards control particularly for measures from higher order IC and MGB, although I agree that many of the other comparisons give similar outcomes for both comparison types. Some comment from the authors on this difference is warranted.

With regard to data extraction and statistical analysis, the raw data (spike rates and LFPs) are extensively massaged before analysis including a normalization procedure but this is standard with these sorts of data. The methods used are presented in a great deal of detail however, facilitating replication of the techniques used. It is interesting that iPE is affected by not only the intensity of the stimulus (larger with low level intensities) but also larger with ascending deviants. Such effects in rodents have also been reported by other research groups recording from epidural or skull recordings in both awake and anaesthetised animals but only limited evidence of a similar effect in humans. The dependence of iPE on intensity is not surprising but the asymmetry with respect to the direction of the deviant (ascending or descending) is puzzling and deserves more attention in the future.

The figures presented in the manuscript are brilliant and exhibit creative ways of demonstrating complex effects at different way-stations in the ascending pathway (eg Figure 3). There were occasions when the lack of a time scale on each plot in a figure was frustrating eg on spike count and firing rate plots of Figure 2 and all of the plots in Figure 5. And there are some instances where it took considerable time to determine what was being presented. For example, in Figure 1, it took some time to determine what the blue and yellow striped sections in Figure 1e depict – it clearly represents the difference between DEV and CTR when CTR is larger than DEV – but it is the choice of colours that puzzled me. Are the authors trying to say that this effect is a combination of prediction error (yellow) and repetition suppression (blue)? The manuscript text and figure caption though suggests that if anything such an effect represents greater cross-adaptation in the DEV than the CTR response. In Figure 5, because the IPE scale on the right hand side of the figure is so indistinct, it took some time to determine what was being depicted in the purple and orange traces in the third segment of the figures. There are some puzzling differences between the timing of effects in the iPE (derived from spike counts) and PEP (derived from LFP) that deserve comment, for instance, at fo-MGB over later latencies but in general there is good correspondence between the two. I am not so convinced that the time course of the PEP effects observed here is comparable to epidural recordings in similar stimulus protocols in rodents but I would agree that the latter are very variable and probably reflect differences in recording protocols (locations of active sensors and reference) and arousal state of the animal (awake vs anaesthetised).

Overall, although I found this a challenging manuscript to read, the data are compelling and the methods exemplary.

Reviewers' comments:

Reviewer #1 (Remarks to the Author):

NietoDiego and colleagues investigate the neural correlates of mismatch negativity. I would recommend publication. The results are topical and interesting

Thank you very much for your supportive comments, we really appreciate them.

my main reservation is the presentation and the pitch of the results. This is detailed in the following:

1. Please rephrase abstract, introduction and discussion. Overall the text lacks a clearly structure introduction with a comprehensive review of previous results (e.g. the results of Taaseh et al. 2011, or Hershenhoren 2014, that both use very similar methods in A1 and come to different conclusions are not discussed). Conversely, in relation to interpretation of SSA as a predictive mechanism, the problem is that the authors imply throughout the manuscript that predictive coding is only used to describe MMN effects (e.g. line 29ff: “we demonstrate that the predictive activity of sensory systems is detectable at the neuronal level” – or line 228: “evidence, hitherto unavailable”)? Single neuron prediction error recordings are quite abundant, and “predictive activity” has been demonstrated in dopaminergic neurons (e.g. Schultz et al. 1997), the cerebellum (e.g. Ohmae & Medina 2015), auditory feedback monitoring in auditory cortex (e.g. Eliades & Wang 2008, Keller & Hahnloser 2009, Schneider et al. 2014), deviance detection in S1 (Musall et al. 2015), and even mismatch processing in the visual system (Meyer & Olson, Zmarz & Keller 2016 – as the authors highlight in the discussion, but also Rao & Ballard 1999, Keller et al. 2012, Saleem et al. 2013, or Roth et al. 2016 for thalamic prediction error signals).

We have rephrased and edited abstract, introduction and discussion significantly, following Reviewer#1 comments. References to all these studies have been implemented, and the results of Nelken and colleagues are now discussed more explicitly. Also, we have emphasized that our results demonstrate the link between predictive coding and MMN at the neuronal level, making it clear that predictive activity in single neurons is already well documented. We are thankful to Reviewer#1 for suggesting this important clarification.

The authors could structure their introduction to highlight that the concepts of predictive coding are quite successful and frequently used to describe brain function, and that these same concepts can also be applied to describe SSA or MMN (as has also been argued previously).

This very helpful suggestion has facilitated the task, thank you. We hope the introduction is more clearly structured now.

Also, in many places the clarity of the manuscript could probably be improved, e.g.:

Line 47f “seamlessly fitting it as the sum of thousands of neuronal predictions error signals”: not only are there grammatical problems in this sentence (e.g. what does ‘it’ refer to), but what the authors are trying to say is unclear (possibly the latter is a consequence of the former)?

We have removed this sentence entirely; we agree it was obscure and not particularly informative.

“higher stations” is likely better described as “higher levels” (of the processing hierarchy).

Certainly, it is changed now in page 3 line 55.

Line 54ff: “neuronal response progressively switch from representing the stimulus itself to represent the sensory prediction error”. Although I think I understand what the authors mean here, this simplification is wrong. In predictive coding, top down signals encode predictions, feedforward signals encode prediction errors, there is no “switching” or even “representation of the stimulus” (technically, everywhere but in the first level of the hierarchy, topdown signals predict the feedforward prediction error, not a stimulus representation in the classical sense)?

This is a really clever observation that can only come from an expert on Predictive Coding. Since we are none, we have not tried to dissect the principles of predictive coding in the introduction; This sentence (being ambiguous if not plain wrong) has been removed and now we just sketch the two most important concepts of predictive coding concerning our research: predictions are sent down, and prediction errors are forwarded up the hierarchy.

Line 55ff: “This is why...” is a bit strong here given that the authors describe a theory.

Removed. Thanks.

Line 87f: “the role of subcortical structures in cognition [...] remains largely unexplored”. Absolutely correct, but irrelevant given that also this manuscript does not address the role of subcortical structures in cognition.

Certainly. This sentence has been removed. However, we meant “perception” instead of “cognition”, as in line 307.

Line 142f: seems unnecessarily aggressive.

We apologize, we didn't mean to sound aggressive at all, only to make the important clarification that the term “SSA” refers to a particular mechanism--instead of just the observed phenomenon--and thus we have replaced it by “neuronal mismatch (nMM)”, and reserve “SSA” to refer to a specific reduction in response to a repetitive tone. We have rephrased the sentence to highlight that this observation concerning nomenclature comes from the very discoverers of SSA (Nelken and Ulanovsky 2007, Taaseh et al 2011), so they cannot be offended by it. And indeed we didn't meant to. So many thanks for drawing our attention to this issue.

2. If the authors want to highlight the relevance of their own data to MMN experiments in humans, they should add a thorough discussion of the influence of the anesthesia used in their experiments.

Thanks very much for this comment. Following Reviewers #2 and #3 as well as the Editor's suggestions, and given our previous experience on awake mice recordings (Duque and Malmierca, 2015; Ayala et al., 2016), we went further and we now include data from new additional experiments on awake mice that clearly replicates the results in the anesthetized condition. Thus, we have demonstrated that anesthesia is not creating any artifactual illusion of deviance detection. This is now implemented in results and discussion.

3. *There are a few apparent inconsistencies in the figures/tables:*

Figure 3b: in the panel the comparison is labeled as n.s., but in table 1 it is reported as $p=0.019$ with a negative change. This just means the DEV response is significantly smaller than the CAS response, not that it is not significant?

Reviewer#1 is absolutely right. Thanks. We wanted to emphasize that the DEV condition was NOT greater than the CAS condition (absence of deviance detection). But of course, the DEV < CAS comparison is actually significant, as stated in Table 1. The corresponding part of Figure 3 has now been removed after Reviewer#1's comment (10), so there is no inconsistency now.

Figure 3e: all neurons/tones should appear in all three "walls" – this is not the case here. In addition, it is very hard to believe – based on the data shown in lower right wall (DEV<CAS) that the distribution of points is on the right of the diagonal (i.e. stronger for the DEV, as table 1 would indicate)?

That is a good observation, indeed. The problem was that the axes limits had been adjusted to 3 spikes in all scatterplots, so a few outliers were not appearing in *some* of the walls.

The DEV > CAS comparison is, indeed, significant (Friedman test between DEV, STD and CAS conditions, with post-hoc comparisons), but it is difficult to see this by the naked eye because many of the points lie together close to the origin; this is why the normalization procedure is so important, to reveal the population behaviour in relative terms.

At any rate, we are not including this part of the figure anymore, since it was difficult to read and did not provide any extra information.

Figure 5c: there is almost no visible difference in the LFP DEV and CAS responses during the first negative peak, but the PEP is clearly negative?

Reviewer#1's impression is, as was ours, that the apparent difference between DEV and CAS traces (for panels c and f) did not correspond to the plotted difference. The reason is that the difference between the negative peaks of the DEV and CAS traces is in fact smaller than the peak of the difference. And the reason for this, in turn, is that the maximal difference between DEV and CAS conditions is not reached by their peaks, but during the upwards slope of both traces. This is further confounded by the thickness of the lines, and by the fact that the differences are relatively small. To illustrate the problem and to convince the reviewer that there is no real discrepancy, we have prepared this detail of Figure 5f, indicating where the max difference is reached (the difference here is around -10 points, corresponding to the negative peak of PEP):

We just made the traces a little thinner, so that this difference is easier to appreciate. We don't include this explanation or indication lines as above in the manuscript figure, unless Reviewer#1 considers it necessary.

We have also changed the way we test for differences in the LFP and iPE traces in Fig. 5. We used to correct for multiple comparisons using the Bonferroni adjustment, but now we use the more powerful false discovery rate method (FDR) from Benjamini and Hockberg. This way, the second significant interval of PEP in Fig. 5f is not broken in two any more.

In addition, in most figures there are axis and line labels missing (e.g. Figure 2 yaxis has only one number, Figure 4a xaxis label is missing, etc. etc.)

We thank Reviewer#1 for her/his detailed observations, we have put much care into the figures now, we hope everything is clear.

Minor suggestions for improvement:

4. Line 43ff: This is misleading at best a biomarker with a d' of roughly 1 (see e.g. Light and Swedlow 2015) can hardly be referred to as a "reliable" biomarker (d' of 1 means e.g. at a 84% hit rate a 50% false positive rate). Please remove "reliable".

Indeed. Thank you.

5. Line 175: please either add the IC terms or explain why you are not listing them. Also, use of only "ho" in the equation (without using "fo") could be considered confusing.

The equation is, indeed, confusing, because of the constant term 0.012 in the model:

$$\text{iPE} = 0.012 + 0.020 \cdot \text{ho} - 0.136 \cdot \text{MGB} + 0.092 \cdot \text{AC} + 0.185 \cdot \text{ho} \cdot \text{MGB} + 0.158 \cdot \text{ho} \cdot \text{AC}$$

This constant actually represents mean iPE in fo-IC. We chose “fo” and “IC” as the reference levels for the “hierarchy” and “nucleus” factors, respectively, and the fitlm in Matlab computes an equation as shown above. We cannot write

$$\text{iPE} = 0.012 * \text{fo} * \text{IC} + \dots$$

instead, that would not be correct, since the 0.012 must be added to all stations, to compute its mean iPE. For example, mean iPE in ho-MGB is

$$\text{iPE}(\text{ho-MGB}) = 0.012 \text{ (base level)} + 0.020 \text{ (because it is ho)} - 0.136 \text{ (because it is MGB)} + 0.185 \text{ (ho-MGB interaction)} = 0.081$$

whereas the mean iPE in fo-IC is just

$$\text{iPE}(\text{fo-IC}) = 0.012 \text{ (base level)}$$

(please note these are means, not medians, as reported elsewhere in the manuscript)

We explain this succinctly when presenting the model equation.

Please note also that fo- and ho- has now been changed to L and NL, according to Reviewer#3's comments. However, in these answers we refer to the initially submitted version of the MS.

6. Line 179ff: Finding a significant effect of one of the variables in a model fit does not imply that all comparisons along that variable are significant. Please rephrase.

Good point. Thanks once more. We rephrased our argument, using also the fact that iPE is significant by itself in ho-IC (Table 1), and including specific post-hoc comparisons to test strict relationships between stations (e.g. to confirm that iPE is statistically higher in ho-AC than in fo-AC, etc).

7. Line 188ff: Why is there an asymmetry between ascending and descending deviants?

That is probably the most intriguing of our observations in this data set; we don't have a convincing hypothesis as to why ascending tones are more efficient to elicit deviance responses, but it might have to do with the spectrotemporal structure of species-specific vocalizations, such that upwards frequency sweeps resemble a puppy call, or something along those lines. We only discuss the fact that similar asymmetries have been observed in humans, but they did not give a reason either.

8. Line 362: “surgical plane of anesthesia” should probably be “surgical stage of anesthesia” (typically, the surgical stage is divided into four planes)?

We are sorry for this confusion. It has been changed to “stage”.

9. Line 683: Figure 1e. I would suggest to label the three different types of differences as DEVCTR (instead of prediction error), STDCTR (instead of repetition suppression), etc. The authors could then state that under the assumption of the predictive coding hypothesis, DEVCTR would correspond to prediction error, etc. The problem is that the schematic currently implies the existence of negative prediction error.

We agree that is an a-priori assumption. We changed that in the figure and explain it better in the figure legend.

10. Figure 3: I suggest either showing the raw data or the normalized data, but not both

We ourselves were actually worried about the complexity of this figure. We now show the normalized data, and associated index histograms, and leave out the raw spike data. As explained earlier, these “raw data” panels do not really provide important new information, and the normalized panels are much easier to interpret and more intuitive. We explain how the normalized panels are constructed in Fig. S1. We are very thankful to Reviewer#1 for this important suggestion, which has helped improved the readability of figure 3.

Reviewer #2 (Remarks to the Author):

Summary

The manuscript by NietoDiego et al. present an interesting body of findings on how novel or deviating stimuli are encoded and detected along the auditory pathway on the level of individual neurons in a predictive coding framework.

To approach this question they recorded extracellular activity in different auditory centres (inferior colliculus IC, thalamus MGB and auditory cortex AC) and presented oddball stimulus sequences to the anesthetized rat. In order to disentangle rather simple adaptive processes (e.g. stimulus-specific adaptation, repetition suppression) from predictive activity (or true deviance detection) they use different carefully designed control conditions. These conditions allow controlling for different effects such as the rarity of a stimulus and regularity of a sequence. Neurons exhibiting predictive activity should respond to a deviating stimulus embedded in an oddball sequence with increased firing rate but with a lower activity to the same stimulus when presented in one of the control conditions.

The central finding of their work is that there is increase in predictive activity along the ascending auditory pathway, even in subcortical structures. This is the first time that predictive single neuron activity has been systematically tracked and demonstrated along an ascending sensory pathway by employing a set of rigorous controls that are essential for a convincing proof. Interestingly, predictive activity evolves gradually along the different stages implying that each auditory centre contributes to its computation or at least amplification. Interestingly, even higher order subcortical structure exhibit predictive activity.

This manuscript is a very comprehensive and important body of work which uses the appropriate methodology and analysis. The work will be of interest to a wider audience, in particular one that is focusing on predictive coding and mismatch negativity (MMN) in human subjects. The understanding of the neuronal basis of MMN is still very superficial despite its extensive usage in studying central sensory processing and pathologies in humans. Due to its comprehensive nature, this paper will certainly be extremely helpful for a broad readership

Thank you very much for your supportive comments, we really appreciate it.

Major comments

Having said this, I have a couple of concerns regarding the figures, methods, results and conclusions.

1. Figures: In general, the figures are not very carefully made. While I understand the urge to make figures concise, it leaves the reader guessing what the axes and scales exactly are. I highly recommend reworking carefully every figure. Just some (incomplete) examples:

Figure 2a: only some x and y axis are labelled properly.

Figure 2b: there are no axis or tick labels at all. This is just not acceptable. Even the bar indicating the stimulus in the last row is missing. There is more than enough space to label them properly

Figure 3 af: There are no ticklabels at all for the scatterplot, is it ranging from 1 to 1?

These scatterplots have been removed from the figure, after Reviewer#1's suggestion. Among other things, as explained above, they were not particularly informative and it was difficult to label them properly.

Figure 4 a, b and c. Only in panel b the first order (fo) and higher order (ho) is indicated for the inferior colliculus. Is it the same order to the other areas? This has to be labelled properly. In panel d the yaxis is not labelled.

Figure 5: No time axis label at all in any panel. Most ylabels are missing as well.

We have reworked every figure carefully to a final stage, we hope everything is clear and complete now.

2. Methods: The authors claim to provide data from single units. However, the only proof of this claim is the spike waveforms in figure 1A and figure 2. This is not enough. The spike detection threshold was set to 23 standarddeviations of the noise floor (page 17, line 25). This is a comparatively low threshold and combined with rather low electrode impedance (1-4 MegaOhm, page 17,line 15) this will typically yield multiunit activity and not single units.

Therefore, measures of the spike quality (distribution of interspikeintervals, real signalnoise ratio, amplitude in μV , Mahalanobis distance, etc. etc.) have to be provided and ideally spikesorting (even without tetrode) should be performed in order convince the reader.

We attempted to use MClust software for Matlab to provide objective clustering quality measures. The problem is that the automatic clustering algorithm (either KlustaKwiq or others that it provides) tends to identify more than one cluster per neuron, which then have to be merged manually. Since we were very certain that our recordings correspond to single units, we provide measures of single-neuron spike clusters for all of our recordings. These are shown now in Figure S3, and corresponding numerical results are reported in the Methods section (page 23).

This leads to another comment that may improve the quality of the manuscript. When looking at the spike waveforms in figure 2A, third row (insets) it seems that there are significant differences in spikewidth between units. So probably, it makes sense to analyse the data with respect to the spike width (fastvs. regular spiking). While it is debatable, if the spikewidth really corresponds to inhibitory or excitatory neurons it would be very interesting to check and see if the findings of Chen, Helmchen & Lütcke 2015 (Journal of Neuroscience) can be confirmed for some auditory areas.

We are currently planning to record intracellularly from the same animal preparations, and we will approach this absolutely relevant question with the appropriate tools.

However, we may include the kind of analysis suggested by Reviewer#2 to complement that study. We hope you understand that we keep that analysis for a future manuscript.

3. Methods: The anaesthesia was set to “moderately deep” (page 16, line 7) for the recordings but “deeply anaesthetized” prior to surgery with urethane (page 12, line 19). I wonder how the authors were able to switch from deep to moderately deep anaesthesia with urethane which is very difficult to control. In general, anaesthesia is a concern. Previous studies in awake animals (e.g. von der Behrens et al. 2009, Journal of Neuroscience; Farley et al. 2010, Journal of Neuroscience) did not find a dramatic change in the effect size of stimulus-specific adaptation compared to similar studies in anaesthetized animals (e.g. Taaseh et al. 2011, Plos One) where the SSA effect seems to be slightly stronger. Furthermore, the typical predictive signal recorded in humans as MMN can be measured under anaesthesia as well. However, Harms et al. 2011 (Front Psychol) saw differences in MMN signals in the rat under anaesthesia compared to the awake condition. And given the fact that the authors are carefully disentangling effects of simple adaptation and real predictive signals it could be very interesting to have a look at the awake condition as well. This would certainly strengthen the manuscript significantly.

Indeed. We are thankful to Reviewer#2 for suggesting this crucial improvement. It has taken time but we were able to record a significant sample from awake mice to confirm this pattern of results, not only of an awake animal, but of a different species. Now we think the validity of our results is far beyond question, and the quality of the manuscript has improved to a new level. These additional results are reported as a new, last section of the Results.

4. Methods: Histology. Apparently, the authors did a histological confirmation of their recordings sites (page 19 ff). However, the proof of this is lacking. This is even more important, as for some regions it is very difficult to draw boundaries between different regions in the subcortical structures. I recommend providing at least examples of histological preparations together with boundaries as supplementary material.

This is now provided in supplementary figures S4 and S5.

5. Methods: The stimulus repetition rates were different for different brain regions (AC: 3 Hz, IC & MGB: 4Hz, page 14, line 2). This is problematic, as it is well known that effects of stimulus-specific adaptation, repetition suppression and predictive coding depend massively on the repetition rate. Therefore, the effect size in cortex is most likely underestimated compared to the subcortical structures (or vice versa). I would recommend choosing just the higher repetition rate (4 Hz) so that each structure is properly driven by the stimulus. The whole evolution of the effect should become clearly by employing this strategy.

We used this configuration of stimulus presentation rates for the reasons explained in the Methods, but we agree that the same rate across stations would provide a more accurate picture of the progression. Nevertheless, we have replicated these results in a different preparation (awake mouse) and using the same presentation rate for AC and IC (4 Hz). Thus, it does not seem to make a significant difference, and we hope this possible flaw of the design would be of no concern. Indeed, as Reviewer#2 suggests, the fact that we used a slower rate in the cortex, actually suggests that if we would have used the faster 4Hz repetition rate the contrast between cortical and subcortical iPE would be even larger. We will certainly explore these kinds of parametric dependences in following detailed studies.

6. *Methods:* As one can see in figure 2, first row, the tuning curves of different units vary significantly. It is known that tuning curves differ between different auditory areas. In general, the tuning in the central nucleus of the IC is sharper than for example in the thalamus or cortex. This leads to two questions. First, the authors should provide measures of the tuning curves in the different structures (in particular for the width Q10dB or Q20dB). This is important because for broadly tuned neurons (as in AC or MGB) there are more stimuli of the control sequences (CAS and MAS) within the tuning curve than for narrowly tuned neurons in the IC where probably only one or two stimuli of the cascade fall within the tuning curve (see figure 2A, first row). The key measure for predictive activity is the response difference between tones embedded in an oddball sequence compared to the same tone embedded in a cascade sequence (CAS). When there is less crossfrequency adaptation in the cascade condition for the IC units one may expect only a small difference between the oddball deviant and the cascade. Currently, the authors do not find predictive activity in the first order IC. However, when they customize the control CAS sequences to the tuning curve width, as they already do for the frequency and amplitude, they may find a different result. At the very least they should look for the correlation between tuning curve width and effect size.

We discuss this issue now in the Results (p 11), analyzing the dependence of iPE on tuning bandwidth. We could not find any correlation, but we observed an interesting trend: neurons with disorganized, fragmented FRAs (which could not be assigned a Q30 value proper) tended to have the highest levels of deviance detection (iPE).

7. *General:* A central finding of the manuscript is the subcortical existence of real predictive activity: “[...] and that this predictive activity emerges hierarchically from subcortical structures” (page 4, line 910, see also page 2, line 9, page 8, line 34, page 8, line 1720 and page 10, line 16). However, according to figure 4B (yellow bars) and figure 5D & E, predictive activity can be only observed in higher order subcortical structures (hoMGB and very weakly in hoIC). These structures receive massive corticofugal projections. Therefore, it is quite misleading to suggest that predictive activity emerges hierarchically from subcortical structures. It is a very likely scenario that the predictive activity is actually projected back from the auditory cortex to MGB and IC. This should be clearly stated in the discussion and in general toned down.

We cannot exclude a cortical influence through top-down projections can modulate or even create the prediction error signals observed subcortically, and we do not mean to suggest that our data demonstrates that they originate subcortically. We have rephrased the MS to highlight the hierarchical organization of prediction error, avoiding terms such as “emergence” that may suggest a subcortical generation. We explicitly discuss this issue now along with anesthesia (line 363 page 16).

8. *Discussion:* The authors find an interesting asymmetry between the ascending and descending cascade stimulus sequence (CAS, figure 4C). In the discussion (page 12, line 45) the authors state that “asymmetries in the direction of frequency change detection (ascending vs. descending) have also been found in both animal (Harms et al. 2014) and human MMN studies (Peter et al. 2010)”. In contrast to the two cited ERP studies, Klein et al 2014 (Brain Topog.) find in their awake single and multiunit recordings only a weak asymmetry of cortical stimulus specific adaptation (SSA) to frequency modulated tones (FM) which can be comparable to the cascade stimulus sequence used in the current manuscript. This should be taken into account.

This closely related result is now appropriately cited and discussed. Thank you.

9. *Figure 3 A1: The 3D projections (insets) are very small and hard to read. I don't think they are particularly helpful for understanding the message of this figure. They just complicate the figure unnecessarily. Panel G1: The key information is in the histograms for iRS, iMM and iPE. Therefore I would just show the histograms for the three conditions and improve their readability. For example one could plot the histograms for one condition (e.g. iPE) below each other following the brain hierarchy. This will facilitate the comparability between distributions of different areas (for one condition) and therefore illustrate the main claim of the manuscript even better ("Hierarchical Prediction Error")*

We did just that, and the readability of the figure has improved significantly. We are thankful to Reviewer#2 for this important suggestion. The mentioned 3D projections appear now only in Fig. S1 to illustrate the normalization procedure.

10. *Discussion: The authors claim that "Previous attempts to show predictive activity in auditory neurons were inconclusive" (page 11, line 56). However, in the next sentence the authors cite two papers that demonstrate exactly this for the auditory and barrel cortex. Furthermore, the authors do not cite a paper by Hamm & Yuste 2015 (Cell reports) who find MMNlike activity in the visual cortex of the awake mouse. This is important because Hamm & Yuste employed the many standards control (MAS) as well demonstrating MMNlike activity in the LFPs and single neurons (2p Calcium imaging) and that it was dependent on the somatostatincontaining interneurons. Therefore, the literature addressing predictive activity in single neurons is not inconclusive anymore. In particular as the authors themselves do not see a big difference between the many standards control MAS (used in the three references mentioned above) and the cascade control CAS.*

The two very relevant works of Musall et al. (2015), Hamm and Yuste (2016), and the newly published Vinken et al. (2017) are now properly discussed (p. 15-16). Also, in the introduction now we emphasize the fact that predictive activity in single neurons is already well established.

Minor comments

1. *Page 13, line 5: "relays" instead of "relies".*

Thank you.

2. *References to figure panels are in uppercase but the panels themselves are written in lowercase. This should be done uniformly.*

We apologize for that inconsistency, we have changed it now.

3. *Figure 4B: what is the interpretation of the negative prediction error (hashed area) in the first order MGB?*

This is a relevant question. We discussed it very briefly but now we have extended that part (around line 356) to highlight the contrast between first- and higher-order MGB with regard to deviance detection. As noted by Reviewer#1, a negative prediction error cannot really exist (or be interpreted that way). IF the iPE index is significantly positive, then we have prediction error. IF iPE is zero, then we don't have response modulation (such as in fo-IC), and IF iPE is negative, then we have cross-frequency adaptation to the DEV tone (this is also sketched in Fig. 1e).

4. *Methods, page 20, line 2223: “[...] to avoid negative response values”. Why were negative values excluded? In particular cortical neurons exhibit a lot of inhibition during sustained stimulation, even during the 75 ms stimulus duration used here. I don’t see the reason to exclude these periods. And was the excitatory response calculated for the time window 0 to 250 ms? On the other hand according to figure legend 2, second row (2) the activity was averaged for all units during 180 ms. A blanket averaging approach for all conditions and brain areas could skew the result significantly as well. This whole procedure should be explained in more detail and the authors have to take into account that neurons in different brain areas exhibit different activity patterns.*

We understand that inhibition is an important response property to be analyzed in any neurophysiology experiment. However, our analysis is rather complex, and it is difficult to interpret negative (inhibited) responses in the context of deviance detection and predictive coding. Positive and negative responses mixed together could lead to values of the indices very difficult to interpret, and could obscure the general pattern of results. We only observed a few inhibited responses in our data set, however, and they were not easy to interpret. Thus, we only included neurons with a significant excitatory response to at least one of the conditions. We have rephrased and explained this part better in the Methods (line 621).

5. *Discussion, page 12, line 6: “[...] demonstrate that prediction error is an intrinsic component of responses of single auditory neurons”. What do the authors mean by “intrinsic” in this context? A prediction error activity is very unlikely the result of a solely intracellular process but more likely the result of network activity. This should be clarified.*

We regret the misunderstanding; it was a poor choice of term. We did not mean “intrinsic” to refer to intrinsic membrane properties of the cell, but to a very fundamental response property of the neurons. We have replaced “intrinsic” with “fundamental”.

6. *Figure 3G and figure 4B: in figure 3G the median iPE is slightly negative (yellow line at yellow distribution for foIC). However, in Figure 4B, the iPE for foIC is slightly positive. Where does this mismatch come from?*

We understand this discrepancy is certainly misleading. We are using median values all through the manuscript (for normalized responses and indices), *except* for the linear model (line 219 p10 in the original ms), because fitlm returns coefficients related to mean values. Since we wanted figure 4B to match the numerical values shown in the model, we represented mean values in figure 4A,B. The main discrepancy occurred for fo-IC, which has a positive mean (old fig 4B) and a negative median (table 1) (none of them really significant). We have changed Fig. 4 to represent median responses (4a) and median indices (4b), so that there is no longer a discrepancy between Fig. 4 and Table 1, which we agree was a potential source of confusion. We have edited Fig. 4 legend accordingly.

7. *Figure 4 legend: Typo “represented”.*

Corrected, thank you.

8. *Table 1: what does a pvalue of 0.000 correspond to?*

These p-values are rounded to 3 decimal figures, so a p of 0.000 means $p < 0.0005$. This is now clearly stated in Table 1 legend.

Reviewer #3 (Remarks to the Author):

The study by NietoDiego et al. is a highly complex study on various stages of the auditory system in deeply anesthetized rats using extracellular single neuron electrophysiology (as well as local field potentials) and various types of tone sequences for auditory stimulation. The study pursues the ambitious goal to “unify three coexisting views of perceptual deviance detection at different levels of description: neuronal physiology, cognitive neuroscience and the theoretical predictive coding framework”. Put more simply, the study strives to establish an animal model of mismatch negativity (MMN) and to test alternative mechanisms as its basis: predictive coding (or Bayesian inference) versus stimulus-specific adaptation (SSA). The authors conclude that “prediction error is an intrinsic component of responses of single auditory neurons”, including in subcortical centers, and forms the basis of human MMN. They hope that this will lead to an understanding of the cellular basis of schizophrenia.

I found the manuscript extremely hard to read and the arguments quite difficult to follow. The complexity of the study lies not only in its design: Cells were recorded from six different brain structures in 36 animals, and were tested with six different tone sequences consisting of 400 (!) stimuli, after determining the frequency response area of a neuron at different intensities. This must take an enormous amount of time and raises concerns about recording stability.

We would like to thank Reviewer#3 for his/her effort in reading and understanding our manuscript. We agree that the design, results and their analysis are a bit too complex, which makes them difficult to present, so we are very pleased that Reviewer#3 has appreciated the full difficulties involved. The project was indeed very challenging and complex, as the nature of the question was: we tried to contrast two explicative models of deviance detection (SSA vs predictive coding) at different hierarchical levels of the auditory system, taking in full consideration its complex anatomical organization (e.g., lemniscal and non-lemniscal) and applying the several different controls that years of research have devised as necessary to interpret the data appropriately. This said, and in response Reviewer#3 and the rest of reviewers, we have made every effort to clarify the manuscript structure, organization and illustration. We are convinced that we have made a substantial progress in this direction, and we hope the reviewer finds now the complexity of the study justified and the presentation of arguments and data straightforward. Regarding Reviewer#3's concerns about the long recording sessions, we did, indeed, spend a lot of time in the recording sessions. At any rate, we have extensive experience in long recording sessions, that require holding a neuron for several hours. As suggested by Reviewer#2, now we provide measures of spike isolation quality (Fig. S3).

The authors also make a number of implicit theoretical assumptions about the organization of the auditory system (and sensory systems in general), which are not straightforward and are not well explained. Some of these assumptions, e.g. that there are “higher order divisions” of the MGB, are tucked away in the Methods section (p. 13), where they are hard to find. These assumptions do not conform with the current understanding of auditory cortex organization in other species.

We agree with Reviewer#3 in this important observation. Again, please bear in mind that the terminology and concepts involved are complex and difficult to present. Our distinction between “first-order” and “higher-order” divisions, as pertaining to different hierarchical levels of processing, is indeed an a-priori assumption, that other researchers may or may not agree with. For this reason, now we don't refer to “first-order” and “higher-order” any more, and we use instead the more widely accepted distinction between lemniscal and non-lemniscal divisions (within IC, MGB and AC).

This distinction is arguably beyond question, since it is based in anatomical and physiological characteristics, and is therefore an objective factor to classify and analyze our data (Results, around line 106). The link between non-lemniscal divisions and a higher-order level of processing is now reserved for the initial part of the discussion (around line 296), where we suggest that from other author's and our own work, we can state that non-lemniscal divisions represent a higher level of processing, especially relevant for deviance detection (see Figure 2 in Winkler et al., 2009). Finally, we have removed any reference to this issue from the Methods section, not to make the impression that we are trying to hide it from the reader. We hope this new, more objective approach would please Reviewer#3, and that this largely terminological issue has been solved in the best possible way. We feel that these subtle but important changes have certainly improved the logical flow of this study, and we are thus sincerely thankful to Reviewer#3 for suggesting them.

Although the total number of cells tested is above 200, which seems like an adequate number overall, this number dwindles to 36 or less in 5 of the 6 structures tested. These are actually quite small sample sizes (Table 1), rendering statistical conclusions less powerful. One of the main conclusions is based on the statistical difference between "firstorder" and "higherorder" neurons, which might require larger sample sizes for the effects to hold up. Furthermore, for the results from "higher order" structures on each level to be collapsed one has to assume that the neuronal responses recorded in these regions are homogeneous, which is almost certainly not the case (see below).

Reviewer#3 is absolutely right in this point, and we agree that there is a huge heterogeneity in neuronal responses along the auditory pathway, especially in the non-lemniscal regions. However, the data clearly shows that when we look at the specific features of SSA and deviance detection, the results are highly consistent. And we and others (Nelken and colleagues; Gutfreund, etc.) have shown this already in previous publications.

With respect to sample sizes, we have now included a statistical power analysis for iPE, that demonstrates that our samples are above minimum required to obtain a statistical power of 0.8 within all stations. This is now explained in Methods (line 680) and data included in Table 1. The sample size is calculated for the total number of frequencies tested (not of single neurons), since that is the input sample for our analysis (Friedman test for DEV vs STD vs CAS comparisons). The sample sizes have been computed for a t-test, since the `sampsizepwr` function in Matlab does not provide an option for non-parametric tests. Thus, this number is used as an approximation, but since iPE is statistically significant in all stations except in IC_L, the major statistical risk we are taking is that the close-to-zero iPE = -0.002 within IC_L was actually statistically significant but we have failed to detect it. Since an iPE of -0.002 has no practical meaning, we don't think this is really a concern. At any rate, we thank Reviewer#3 for promoting this important aspect of the analysis.

Furthermore, neurons in nonprimary auditory cortex are notoriously hard to drive with pure tones (including presumably puretone sequences), so determination of an FRA may not always be reliably possible, which jeopardizes subsequent analysis.

Reviewer#3 is, once more, absolutely right in this point. Actually, there is a correspondence between neurons with unstructured, fragmented FRAs, and high levels of deviance detection, as included now in our analyses (around line 256 p11). We already noticed that many years ago, when we first studied SSA. However, it should be noted that the FRAs are constructed with a random presentation of different combinations of frequencies and intensities, which prevent adaptation effects. It is not

until we used the oddball and other adaptation paradigms that the responses would be affected by these effects. But this is indeed what we are probing. Moreover, the fact that higher-order neurons are so hard to drive by “meaningless” pure tones, but they respond strongly to pure tones when they are deviant, indicate a sharp preference of these neurons to contextual meaning, rather than physical features of the stimulus. In other words, we show that nonprimary neurons are notoriously hard to drive with pure tones, *unless these tones are deviants from a regularity* (Fig. 2f).

One could also argue that recordings in anesthetized animals make the results relatively less convincing, since the ultimate conclusions are claimed to be about basic mechanisms of cognition. Recordings in awake animals would seem to be preferable.

Indeed. And following this Reviewer's, as well as Reviewer#2's and the Editor's comments, we have now made additional experiments in awake mice that confirm and replicate our results in anesthetized rats. We are content that these new experiments show that our results are a fundamental property independent of the animal's state. Please see also responses to Reviewer#2 comments.

Another possible shortcoming is that, although the authors performed histological reconstructions of their electrode tracks, which were used to assign the recorded cells to different structures, they did not report recording depths of their cortical neurons. Thus they are unable to provide information on layer specific effects in auditory cortex, which may be important, as there is classical evidence that neurons in supragranular and infragranular layers of cortex play different roles in feedforward and feedback processing (Van Essen; Rockland & Pandya; Bullier & Kennedy; and others). Admittedly, the thinner extent of cortex in rodents than in primates makes it harder to determine laminar specificity.

We agree with Reviewer#3, and in fact we analyzed the correlation between cortical depth (as measured with our microdrive device) and iPE, but we found no differences across depth; as noted by Reviewer#3, probably due to the thinner extent of cortex in the rat, and the intrinsic inaccuracy of depth values as obtained from the microdrive (due to tissue indentation, not perfect straight angle of penetration, etc). However, in the near future we aim to study anatomical relations, not only with regard to specific nuclei, but also as layer-specific effects, using in-vivo patch clamp recordings. This will be very challenging and time consuming, but hopefully in the near future we will be able to answer this extremely important question.

The authors reference the studies of Lee and Sherman (2011) as their model for the organization of ascending pathways into “first order” and “higher order” structures at every level. According to this model, different subnuclei in the inferior colliculus (IC) and the medial geniculate body (MGB) are at different hierarchical levels, hence the terms “first order” and “higher order”. A competing notion is that these subnuclei are organized in parallel and mediate different functions. In the barn owl, for instance, a central nucleus (ICC) is responsible for tonal or frequency information, whereas an external nucleus (ICX) is important for spatial hearing. The same assumption of functional subdivisions and parallel processing streams has been made for the auditory system of higher mammals, including humans and nonhuman primates (Kaas, Hackett, and others): Although the overall organization is obviously hierarchical from brainstem to cortex, there are multiple parallel stages at each level performing different functions, beginning with a dual architecture at the level of the cochlear nucleus (VCN and DCN). This parallel hierarchical organization is epitomized at the cortical level, where multiple functionally specialized areas are assumed to work in parallel (in addition to forming

several hierarchical levels). Multiple primarylike areas exist as well. The assumption of just one “information bearing” structure at every level (“fo”) and one higherorder (modulatory) structure (“ho”) seems too simplistic.

The Reviewer comments are interesting and we are not opposed to his/her views. As explained above, now we refer to lemniscal and nonlemniscal divisions of auditory centers. We also highlight in the discussion that, as correctly indicated by Reviewer#3, lemniscal and nonlemniscal divisions represent two parallel pathways for processing of different properties of sound. However, we also highlight the hierarchical organization of the auditory pathway, from IC to AC, as well as from lemniscal to non-lemniscal, which is also hierarchical, since primary centers tend to code for the physical features of stimuli, whereas nonlemniscal centers tend to have wider receptive fields and be more sensitive to the context in which stimuli are presented (Malmierca et al., 2009; Antunes et al., 2010; Lee and Sherman, 2011; Duque et al., 2012; Atiani et al., 2014; Lee, 2015; Nieto-Diego and Malmierca, 2016). We have included additional discussion sections along these lines comparing the organization of the visual and auditory system as it has been previously suggested that the computational capacity of the IC in the auditory system would be equivalent to that of the primary visual cortex (King and Nelken, 2009).

Having said that, we agree that our analysis involves important simplifications of the complex reality regarding the organization of the auditory pathway. We are entirely aware of these simplifications, but our main concern was to make these results, already complex enough, as easy to follow as possible.

In summary, it would be more effective to concentrate on the most interesting finding which, in my opinion, is that auditory cortex is in fact substantially different from subcortical structures (see Fig. 3 c,f) (though the authors emphasize a different view), and that nonprimary auditory cortex, in particular, is different from all other structures in that it shows by far the clearest signs of predictive coding (Figs. 4b,d and 5f). Whether this is indeed the neural correlate of MMN (and whether this has any relevance for schizophrenia) remains to be seen until the electrophysiological findings can be compared with behavioral results in the same species. More specificity for different brain regions in, e.g., temporal and frontal cortex would also have to be demonstrated.

Reviewer#3 is right, and we agree with his/her view, in that the auditory cortex is the most interesting result here, but we can conclude that precisely because we have the midbrain and thalamus data to compare with. This allowed us to understand the transformation that occurs from midbrain to thalamus to cortex, and to show the clear hierarchical organization of prediction error along the pathway. Also, the fact that neuronal prediction error can be found at subcortical levels, is arguably more novel than the cortical results and is being now started to be taken into account (Aukstulewicz and Friston, 2016). So our data adds the necessary empirical basis to support this conceptually important notion.

Reviewer #4 (Remarks to the Author):

NietoDiego et al provide evidence of a hierarchical organisation of prediction errors at a neuronal level in the ascending auditory system: they observed minor levels of predictive processing in IC, but in higher order IC areas only, similarly in MGB but at an increased level relative to IC and substantial predictive error in AC in both first and higher order AC areas. The importance of this finding is that these observations are at the neuronal level rather than at large scale network level such as exemplified by mismatch negativity (MMN) recorded from scalp recordings and clearly provide evidence of hierarchical organisation of increasing predictive error from subcortical

areas to cortical areas. The design of the study is exemplary in that the authors have used optimal control conditions to separate predictive error effects from repetition suppression or stimulus specific adaptation (SSA) effects.

While I have queries or require clarification about some conclusions or Figures, the findings will be of considerable interest to neuroscientists in general. Predictive coding frameworks are increasingly being adopted as models of brain function following Friston's seminal publications. Repetitive patterns of stimulation are proposed to generate internal models of the environment that enable the brain to anticipate its next state of activation. When sensory input produces an activation pattern that matches model predictions, prediction error is largely suppressed, thus assisting the brain to conserve resources when sound provides no new information. However, change in the environment leads to the elicitation of prediction errors signalling that an update to the model may be required. The somewhat corticocentric literature on predictive coding to date implicates a cortical hierarchy whereby the predictions of the model are expressed in top-down (higher to lower areas) connections that control the responsiveness to sensory input. Responsiveness is reduced in cortical regions encoding the predicted activation (e.g., the neurons sensitive to the attributes of the predicted sound), but it is increased in alternate cortical regions (sensitive to nonpredicted attributes). The discrepancy generates an error signal communicated bottomup (lower to higher areas).

What NietoDiego et al have shown is that predictive error coding begins at midbrain levels. All areas investigated by NietoDiego et al showed evidence of response suppression (or what others have referred to as adaptation or SSA) that is particularly marked at subcortical areas but as one moves up the auditory ascending pathway, predictive error processing becomes dominant. The paper will therefore have a strong influence on the field and lead to further research in subcortical nuclei and the feedforward/ feedback mechanisms that account for critical features of predictive coding.

We are very grateful to Reviewer#4 for his/her supportive words.

As noted earlier, the methodology is exemplary – in terms of stimulus control, responses to same physical stimulus (same intensity and frequency) when it is a standard (SDT) in an oddball sequence or a deviant (DEV) or a control (CTR) stimulus in a cascade (CAS) or manystandards (MAS) sequence are always employed), and the distance between each of these stimulus types is kept fixed at 0.5 octave (in terms of frequency). The use of the cascade control sequences is very clever and confirms my suspicion that the reason the cascade sequence did not work (in that it did not confirm evidence of true deviance detection) in surface recordings in the rat is because the equivalent of the oddball deviants were placed at the extremes of the ascending and descending cascades (see Harms et al., 2014) and therefore probably as deviant as an oddball deviant. NietoDiego et al's decision to place the equivalent of the deviant in the middle of either an ascending or descending sequence (although not really equivalent to the original cascade sequence as used by Ruhnau et al 2012) is sensible – it controls not only for background regularity and probability but also for the frequency (Hz) of the preceding stimulus (which is equivalent to the standard of an oddball sequence).

Of course if rodents are unable to model the regularity (either an ascending or descending sequence of sounds) and we have no data that I know of on this issue, then the cascade sequence presumably acts in much the same way as the many standards control. The authors state that the outcomes of their analyses are much the same whether they use cascade or many standards control sequences for comparison purposes with DEV (or STD) responses but inspection of Table 1 suggests that the cascade control may have provided a more sensitive measure of prediction error (iPE)

than the many standards control particularly for measures from higher order IC and MGB, although I agree that many of the other comparisons give similar outcomes for both comparison types. Some comment from the authors on this difference is warranted.

The comparison between MAS and CAS responses is a very relevant one for the case of predictive coding, since the only difference between these two sequences is that MAS is random, whereas CAS is regular, and thus predictable. However, we do not find significant differences in our sample (line 145). As pointed out by Reviewer#4, the rat brain may be unable to encode for such a complex regularity (CAS), at least at a slow presentation speed, and without previous exposure or learning of the pattern. At 3 stim per second, the whole cascaded pattern takes 3.33 seconds to complete. Maybe with a faster presentation rate, say 8-10 stim/second, the regular pattern of the CAS sequence could be more easily encoded and recognized by the rat brain. We will certainly address this important question in future studies.

However, we do find some interesting differences in neuronal responses to MAS and CAS sequences. In particular, values of the index of prediction error (iPE) computed using the CAS as control are higher and tend to be more significant than using the MAS control, especially at subcortical levels. This may indicate that the regularity present in the CAS condition may be just what is required to boost iPE levels at subcortical structures just enough to make them detectable and significant. We can't, however, speculate further on the issue, until we do additional experiments comparing regular vs irregular mini-sequences.

All these considerations are now discussed starting line 319. We sincerely thank Reviewer#4 for promoting this interesting debate.

With regard to data extraction and statistical analysis, the raw data (spike rates and LFPs) are extensively massaged before analysis including a normalization procedure but this is standard with these sorts of data. The methods used are presented in a great deal of detail however, facilitating replication of the techniques used. It is interesting that iPE is affected by not only the intensity of the stimulus (larger with low level intensities) but also larger with ascending deviants. Such effects in rodents have also been reported by other research groups recording from epidural or skull recordings in both awake and anaesthetised animals but only limited evidence of a similar effect in humans. The dependence of iPE on intensity is not surprising but the asymmetry with respect to the direction of the deviant (ascending or descending) is puzzling and deserves more attention in the future.

Indeed, this asymmetry is a puzzling result, one which cannot fully interpret. Its potential relevance is discussed following line 346 (p15), and a new ref (Klein et al., 2014) has been added after Reviewer#2 suggestion.

The figures presented in the manuscript are brilliant and exhibit creative ways of demonstrating complex effects at different way stations in the ascending pathway (eg Figure 3). There were occasions when the lack of a time scale on each plot in a figure was frustrating eg on spike count and firing rate plots of Figure 2 and all of the plots in Figure 5.

Thank you. It took considerable thought and time to come up with these figures. We apologize for the lack of labeling in some cases. We have revised all figures for mislabeling or lack of labeling, and we hope they read well now.

And there are some instances where it took considerable time to determine what was

being presented. For example, in Figure 1, it took some time to determine what the blue and yellow striped sections in Figure 1e depict – it clearly represents the difference between DEV and CTR when CTR is larger than DEV – but it is the choice of colours that puzzled me. Are the authors trying to say that this effect is a combination of prediction error (yellow) and repetition suppression (blue)? The manuscript text and figure caption though suggests that if anything such an effect represents greater cross adaptation in the DEV than the CTR response.

We understand that the concept of a negative prediction error is contradictory, and thus calling “prediction error” to the DEV-CTR difference can only be done when this difference is positive. Following Reviewer#1’s suggestion, these differences have been now labeled as “DEV-CTR” and “CTR-STD” in Fig. 1d, and then their interpretation as “prediction error” and “repetition suppression” is stated in the figure legend. This way, we think there is no inconsistency now with Fig.1e, that shows these differences without further assumptions. As shown in Fig. 1e, only if DEV-CTR is found positive (as is the case eventually), could we interpret it as prediction error.

In Figure 5, because the iPE scale on the right hand side of the figure is so indistinct, it took some time to determine what was being depicted in the purple and orange traces in the third segment of the figures. There are some puzzling differences between the timing of effects in the iPE (derived from spike counts) and PEP (derived from LFP) that deserve comment, for instance, at fo-MGB over later latencies but in general there is good correspondence between the two. I am not so convinced that the time course of the PEP effects observed here is comparable to epidural recordings in similar stimulus protocols in rodents but I would agree that the latter are very variable and probably reflect differences in recording protocols (locations of active sensors and reference) and arousal state of the animal (awake vs anaesthetised).

We have tried our best to make the PEP and iPE scales more distinct, by bolding or outlining the letters. Discrepancies between PEP and iPE traces at subcortical levels arise from highly noisy signals with random, nonsignificant, fluctuations. For example, the positive iPE values observed in fo-MGB at late latencies are not significant, indicating the absence of an iPE, so there is no discrepancy there.

The treatment of the PEP as an “MMN-analog” signal is of course not strictly correct, since the PEP is recorded from inside the cortex and has a much shorter range. However, there is certainly a relation between this PEP and the real MMN, and this is the basis for our conclusions.

Overall, although I found this a challenging manuscript to read, the data are compelling and the methods exemplary.

We are very thankful to Reviewer#4 for his/her in-depth and positive review. We feel these are important results that might be of interest for a wide audience, but at the same time we acknowledge these are certainly complex results, and we have done our best to simplify them and represent them in the clearest possible way.

Reviewers' comments:

Reviewer #1 (Remarks to the Author):

All my concerns were addressed, and I have no further comments.

Reviewer #2 (Remarks to the Author):

Review of revision 1 "Hierarchical Prediction Error in Neuronal Responses along the Auditory Neuraxis"

The authors reworked the manuscript significantly. The authors took great care to address every concern by the reviewers which they do with great success. Besides new analysis they performed additional experiments. As expected, the awake recordings in mice confirm their findings from the recordings under anaesthesia in rats. Additional information about the cluster quality and histological confirmation of the recording sites is provided as supplementary material.

Overall, the manuscript improved significantly after this revision and I would like to congratulate the authors for their achievements. I think this work should be published as soon as possible the way it is. It is highly original work and addresses an important question in the field of central sensory processing. This paper will be greatly appreciated by a very broad readership and it will be a keystone for a lot of future research by many groups.

Reviewer #3 (Remarks to the Author):

The authors have made a great effort to respond to the dozens of criticisms they received from all four reviewers. I'm afraid to say that they have not accomplished their goal to make the manuscript more understandable. I still have the same difficulty to follow their arguments and to appreciate their conclusions. Not only are they mixing terminologies and are using a multitude of idiosyncratic abbreviations, the figures are still of insufficient quality, and the theoretical framework remains unconvincing.

While it is true that auditory physiology is a very heterogeneous field and people working on different species often seem to live in different universes, it should still be possible to explain the assumptions, results and conclusions of a study to workers in a neighboring field. If the Editors picked me intentionally (as I assume) as a reviewer representing readers from such neighboring fields, the authors certainly did not succeed.

- If, for instance, terminology highly specific to the auditory system (lemniscal and non-lemniscal) is used without further differentiating other nuclei or subnuclei in the thalamus, researchers in the visual system will have a hard time accepting that this model can easily be transferred to their system and could be a generalized model of predictive coding.
- If a derogatory term like 'corticocentric' is used without apparent detailed knowledge of the serial and parallel organization of the cerebral cortex into dozens of functionally

specialized fields, readers working on parietal cortex will not accept that possible alternative models, in which the MMN may be the result of a mismatch between, e.g., frontal and parietal-temporal cortex, are not even discussed and all non-primary (or 'non-lemniscal') areas are equated into a single level. If, furthermore, laminar organization of cortex and its importance for the distinction between feedforward and feedback processing is essentially dismissed, then the model cannot be taken seriously and one might even question whether mice or rats with their thin cortex are the best model for these types of studies.

Since the other reviewers who (judging from their intimate familiarity with the authors' jargon) are closer to the field, don't seem to find any fault with their techniques, I might recommend publication in a specialty journal, as this could form the basis of a very narrow rodent model for mismatch negativity, but certainly not more than that. I am strongly opposed, however, to publication in Nature Communications.

Reviewer #4 (Remarks to the Author):

My assessment of this manuscript has not changed. While the design and analyses of the data are complex, and will challenge the reader, the authors have compelling findings on the evolution of predictive coding in a hierarchical fashion from subcortical areas to cortical and differential involvement of the non-lemniscal (NL) and lemniscal (L) auditory systems. I must admit to also being more comfortable with the NL and L terminology as suggested by other reviewers. The extension of the findings in anaesthetised rats to awake mice (using a more limited coverage of auditory relay stations and a smaller number of animals) is valuable as it confirms that the original results can be generalised to a different species and are not a function of the anaesthetised state of the animal. The paper represents a significant advance to the literature on where predictive coding emerges in the ascending auditory system and will be of great value to both basic and clinical researchers interested in auditory mismatch responses.

My only concern is that the new data on awake mice although limited in nature is not well integrated with the rest of the manuscript. Would the authors consider adding to the last paragraph of their introduction that in this study they will be reporting data from a large sample of anaesthetised rats and from a smaller sample of awake mice to assess the generalisability of any findings across species and arousal state. The first mention of the awake sample is in the results. There is a similar gap in the abstract.

I have only a few other minor comments and suggestions only:

First paragraph of the main text:

The sentence " Brain responses to the perceptual mismatch between expected and actual sensory inputs have been extensively recorded in all sensory systems including auditory 3, visual 4, somatosensory 5 and olfactory 6 modalities, and are thought to underlie the brain's ability to resolve auditory objects" can be read as implying that perceptual mismatches in all sensory systems underlie the brain's ability to resolve auditory objects whereas in fact it is only the auditory system that does so.

Line 150: Change "commensurable" to "commensurate". Although it is not entirely clear to me that using CAS or MAS led to equivalent results – see later comments in the manuscript on the advantages of using CAS particularly in the awake state (see statement on line 321-322 in the discussion and Table 1). It would be more accurate to say the results were largely comparable.

Line 159: The statement "However, as recognized by the latter authors, the...": I recommend replacing latter authors with the names of the authors.

Line 489-490: Report age of mice in weeks as well (8 – 12 weeks?) to be comparable with rats

Line 503: The details of free-field stimulation in the awake animals should be specified in more detail – characteristics of the speaker (frequency response) and the sound levels.

Lines 508-511: It needs to be clarified that what is being described here is how sound was delivered to the anaesthetised rats.

Line 583: Distinct tonotopic maps are described in rats. What about mice?

RESPONSE TO REVIEWERS

Reviewers' comments:

Reviewer #1 (Remarks to the Author):

All my concerns were addressed, and I have no further comments.

Our response: Thanks very much. We are delighted that we could address all issues successfully

Reviewer #2 (Remarks to the Author):

Review of revision 1 "Hierarchical Prediction Error in Neuronal Responses along the Auditory Neuraxis"

The authors reworked the manuscript significantly. The authors took great care to address every concern by the reviewers which they do with great success. Besides new analysis they performed additional experiments. As expected, the awake recordings in mice confirm their findings from the recordings under anaesthesia in rats. Additional information about the cluster quality and histological confirmation of the recording sites is provided as supplementary material.

Overall, the manuscript improved significantly after this revision and I would like to congratulate the authors for their achievements. I think this work should be published as soon as possible the way it is. It is highly original work and addresses an important question in the field of central sensory processing. This paper will be greatly appreciated by a very broad readership and it will be a keystone for a lot of future research by many groups.

Our response: Thanks very much. We are delighted that we could address all issues successfully and appreciate the very nice comments and positive recommendation that the manuscript should be published as soon as possible.

Reviewer #3 (Remarks to the Author):

The authors have made a great effort to respond to the dozens of criticisms they received from all four reviewers. I'm afraid to say that they have not accomplished their goal to make the manuscript more understandable. I still have the same difficulty to follow their arguments and to appreciate their conclusions. Not only are they mixing terminologies and are using a multitude of idiosyncratic abbreviations, the figures are still of insufficient quality, and the theoretical framework remains unconvincing.

While it is true that auditory physiology is a very heterogeneous field and people working on different species often seem to live in different universes, it should still be possible to explain the assumptions, results and conclusions of a study to workers in a neighboring field. If the Editors picked me intentionally (as I assume) as a reviewer representing readers from such neighboring fields, the authors certainly did not succeed.

Our response: Thanks very much for your comments. Once more we appreciate the time and effort made by this reviewer.

We have made enormous additional efforts and tried our best to implement all the ideas and criticisms expressed made by the reviewer. We sincerely hope that this new, and much revised version of the manuscript is clearer and that all explanations and arguments can be easily followed by any reader. Our goal was to make our study useful for as many researchers as possible in all relevant fields.

- If, for instance, terminology highly specific to the auditory system (lemniscal and non-lemniscal) is used without further differentiating other nuclei or subnuclei in the thalamus, researchers in the visual system will have a hard time accepting that this model can easily be transferred to their system and could be a generalized model of predictive coding.

Our response:

We have now added a new section (pages 4-5, lines 90-109) following the editor recommendation and also after consultation with 3 additional colleagues, one being an expert on the auditory system (Dr. Nell Cant), another being an expert in the primate auditory system and also working on MMN (Dr. Yonatan Fishman) and finally a third one being an expert in the primate visual system (Javier Cudeiro). They all agree that the new paragraph we had implemented in the introduction helps to clarify the nomenclature. While the terminology of lemniscal vs. non-lemniscal is common in the auditory and somatosensory system it does not apply to the visual system, but this should not preclude the generalization of model of predictive coding, as in fact it was first conceptualized in the context of visual processing (Rao and Ballard, 1999; Lee and Mumford, 2003).

- If a derogatory term like ‘corticentric’ is used without apparent detailed knowledge of the serial and parallel organization of the cerebral cortex into dozens of functionally specialized fields, readers working on parietal cortex will not accept that possible alternative models, in which the MMN may be the result of a mismatch between, e.g., frontal and parietal-temporal cortex, are not even discussed and all non-primary (or ‘non-lemniscal’) areas are equated into a single level. If, furthermore, laminar organization of cortex and its importance for the distinction between feedforward and feedback processing is essentially dismissed, then the model cannot be taken seriously and one might even question whether mice or rats with their thin cortex are the best model for these types of studies.

Our response: First of all, we sincerely apologize for the use of the term ‘corticentric’. In any case, the term was used only once and we did not plan intentionally to convey a derogatory meaning or connotation, so we thank the reviewer for drawing our attention to this delicate issue. By contrast, we simply wished to emphasize the fact that up till now, actually all the existing literature has not considered, or has even neglected the role of subcortical nuclei in models of stimulus-specific adaptation and predictive coding. To avoid future misunderstandings and/or that any reader working in the cortex is offended we have replaced the term, and hopefully now, the explanation given is more neutral and conciliatory (page 16, line 354).

Thanks very much also for your additional comments. These comments have been very useful and we have further implemented new sections in the discussion to comment on the limitations of our study, including the importance of the laminar organization of the AC (Page 20, lines 422-449), the differential processing between feedforward and feedback processing as well as the relationships between predictive coding, frontal cortex and attention. Although this important issue was outside the scope of our study, we now suggest future studies that will shed light on this. Furthermore, we have remarked on the results of some previous studies and the role of prefrontal cortex in MMN and their possible relation to our results (Page 20, lines 450-458).

We thank the reviewer for this insightful suggestion as we think the arguments put forward in this new section are now more robust as the text flows much better making the narrative more friendly and useful for a broader audience.

We have also commented on the validity of the rodent studies as compared to primates. In this case, we respectfully disagree with this reviewer opinion. In fact, we think we were fortunate to have selected rodents as the animal model because this choice may have enabled us to discover the hierarchical organization of prediction errors. For obvious technical and ethical issues, previous studies in primates and humans used local field potentials, current source density components, multiunit activity, and/or Hy-band responses. These techniques are excellent for population activity, but the activity patterns may be missed if present at a finer neuronal level or in elusive neuronal responses. Our experiments using single unit recordings may have facilitated the discovery and detection of these subtle neuronal responses. This is now discussed on pages 19-20; lines 422-441.

Since the other reviewers who (judging from their intimate familiarity with the authors' jargon) are closer to the field, don't seem to find any fault with their techniques, I might recommend publication in a specialty journal, as this could form the basis of a very narrow rodent model for mismatch negativity, but certainly not more than that. I am strongly opposed, however, to publication in Nature Communications.

Our response: We regret that the reviewer is so strongly opposed to see our paper published in Nature Communications. As the reviewer recognized in his/her first sentence of this second round, we have implemented all the dozens of suggestions and even made new experiments in an awake preparation as suggested by all the reviewers, including reviewer #3. We sincerely value this reviewer's comments because they (together with all other comments from the other 3 reviewers) helped us to make the manuscript better and stronger. Our experiments are well designed, the data is very robust and hopefully the flow of the manuscript is clearer and sharper now. We are convinced that our study will change the way of thinking in the field and serve as the foundation for the design of new and future projects.

Reviewer #4 (Remarks to the Author):

My assessment of this manuscript has not changed. While the design and analyses of the

data are complex, and will challenge the reader, the authors have compelling findings on the evolution of predictive coding in a hierarchical fashion from subcortical areas to cortical and differential involvement of the non-lemniscal (NL) and lemniscal (L) auditory systems. I must admit to also being more comfortable with the NL and L terminology as suggested by other reviewers. The extension of the findings in anaesthetised rats to awake mice (using a more limited coverage of auditory relay stations and a smaller number of animals) is valuable as it confirms that the original results can be generalised to a different species and are not a function of the anaesthetised state of the animal. The paper represents a significant advance to the literature on where predictive coding emerges in the ascending auditory system and will be of great value to both basic and clinical researchers interested in auditory mismatch responses.

Our response: Thanks very much. We are delighted that we could address most issues successfully already and appreciate the very nice comments and positive recommendation. We hope that the new suggestions are now fully addressed.

My only concern is that the new data on awake mice although limited in nature is not well integrated with the rest of the manuscript. Would the authors consider adding to the last paragraph of their introduction that in this study they will be reporting data from a large sample of anaesthetised rats and from a smaller sample of awake mice to assess the generalisability of any findings across species and arousal state. The first mention of the awake sample is in the results. There is a similar gap in the abstract.

Our response: The reviewer is absolutely right, we have now amended it as suggested (please see pages 2, lines 31-32 and page 5, lines 110-111).

I have only a few other minor comments and suggestions only:

First paragraph of the main text:

The sentence “ Brain responses to the perceptual mismatch between expected and actual sensory inputs have been extensively recorded in all sensory systems including auditory 3, visual 4, somatosensory 5 and olfactory 6 modalities, and are thought to underlie the brain’s ability to resolve auditory objects” can be read as implying that perceptual mismatches in all sensory systems underlie the brain’s ability to resolve auditory objects whereas in fact it is only the auditory system that does so.

Our response: Indeed, Fixed now, thanks for the comment. This make the issue more accurate and clear. (please see page 2, lines 43 and 44).

Line 150: Change “commensurable” to “commensurate”. Although it is not entirely clear to me that using CAS or MAS led to equivalent results – see later comments in the manuscript on the advantages of using CAS particularly in the awake state (see statement on line 321-322 in the discussion and Table 1). It would be more accurate to say the results were largely comparable.

Our response: done, please see page 8, lines 182-183.

Line 159: The statement “However, as recognized by the latter authors, the...”: I recommend replacing latter authors with the names of the authors.

Our response: done, please see page. 9, line: 193.

Line 489-490: Report age of mice in weeks as well (8 – 12 weeks?) to be comparable with rats

Our response: done, please see page. 25, line: 576-577.

Line 503: The details of free-field stimulation in the awake animals should be specified in more detail – characteristics of the speaker (frequency response) and the sound levels.

Our response: This was indeed missing, thanks for the observation. It is now explained and detailed on page. 27, lines: 609-617.

Lines 508-511: It needs to be clarified that what is being described here is how sound was delivered to the anaesthetised rats.

Our response: Again, this is a very good point. It is done now, please see page 26, line: 593.

Line 583: Distinct tonotopic maps are described in rats. What about mice?

Our response: Thanks, now it is better detailed, please see page. 30-31, lines: 688-694.